# Testing the Electrodynamic Method to Derive Height-Integrated Ionospheric Conductances

Daniel Weimer[1,2] and Thom Edwards[3]

[1]Center for Space Science and Engineering Research, Virginia Tech, Blacksburg, Virginia, USA
[2]National Institute of Aerospace, Hampton, Virginia, USA
[3]DTU Space, Technical University of Denmark, Copenhagen, Denmark

**Correspondence:** Daniel Weimer (dweimer@vt.edu)

**Abstract.** We have used empirical models for electric potentials and the magnetic fields in both space and on the ground to obtain maps of the height-integrated Pedersen and Hall ionospheric conductivities at high latitudes. This calculation required use of both "curl-free" and "divergence-free" components of the ionospheric currents, with the former obtained from magnetic fields that are used in a model of the field-aligned currents. The second component is from the equivalent current, usually associated with Hall currents, derived from the ground-level magnetic field. Conductances were calculated for varying combinations of the Interplanetary magnetic field (IMF) magnitude and orientation angle, as well as the dipole tilt angle. The results show that reversing the sign of the Y component of the IMF produces substantially different conductivity patterns. The Hall conductivities are largest on the dawn side in the upward, Region 2 field-aligned currents. Low electric field strengths in the Harang discontinuity lead to inconclusive results near midnight. Calculations of the Joule heating, obtained from the electric field and both components of the ionospheric current, are compared with the Poynting flux in space. The maps show some differences, while their integrated totals match to within 1%. Some of the Poynting flux that enters the polar cap is dissipated as Joule heating within the auroral ovals, where the conductivity is greater.

## 1 Introduction

The Earth's ionosphere has a major role in the flow of currents and energy within the magnetosphere, or what is also known as the "geospace environment." The currents in the ionosphere are responsible for geomagnetic effects seen at the Earth's surface, and they also have a role in the high-latitude heating of the upper atmosphere. The magnitude of these effects is determined to a large extent by the level of conductivity in the ionosphere, and as such the conductivity needs to be accurately known for reliable geospace modeling. On the other hand, it can be argued that the conductivity values are not known with a high precision, and may be the least well-quantified part of the coupled magnetosphere-ionosphere system.

This problem is not due to a lack of understanding, as the basic equations that define the conductivity values are known. The reference books by Rees (1989); Prölss and Bird (2004); Brekke (2013) and others provide formulas for calculating the

Pedersen ($\sigma_P$) and Hall ($\sigma_H$) conductivities. While these formulas are basically the same, they are presented in a variety of different formats, symbols, and notations, which can lead to confusion. A version that we find more preferable follows:

$$25 \quad \sigma_P = \frac{n_e\,|e|}{B}\left[\frac{r_e}{1+r_e^2} + \sum_i C_i \frac{r_i}{1+r_i^2}\right] \tag{1}$$

$$\sigma_H = \frac{n_e\,|e|}{B}\left[\frac{1}{1+r_e^2} - \sum_i C_i \frac{1}{1+r_i^2}\right] \tag{2}$$

where $n_e$ is the electron number density, $B$ is the magnitude of the magnetic field, $e$ is the fundamental constant for the charge of an electron, and $C_i$ is the relative ion concentration for the $i$th ion species, that are assumed to have a total number density equal to that of the electrons. The ratio $r_i$ is defined as:

$$30 \quad r_i = \nu_{in}/\Omega_i = 1/k_i \tag{3}$$

where $k_i$ is the "ion mobility coefficient" (Brekke, 2013), $\nu_{in}$ is the ion-neutral or electron-neutral collision frequency, and $\Omega_i$ refers to the cyclotron frequency:

$$\Omega_i = |e|\,B/m_i \tag{4}$$

The $r_e$ ratio is obtained substituting electrons for ions in Eq. (3) and (4). The absolute value of the electron charge is used in
the equations above in order to reduce sign ambiguity. Equations (1) and (2) are similar to Eq. (5) and (6) by Mallinckrodt (1985) (with a sign correction), and simplified using Eq. (3) and (4).

The height-integrated values of these conductivities are often used, designated with upper-case symbols $\Sigma_P$ and $\Sigma_H$. In order to calculate these height-integrated values it is necessary to know the magnetic field strength, electron temperature and number density, and ion and neutral composition and number densities of each species, at all altitudes within the ionosphere.
At low and mid-latitudes these quantities are better known and can be obtained from a reference magnetic field model, and familiar empirical models of the ionosphere and neutral atmosphere such as the "International Reference Ionosphere" (IRI) (Bilitza, 2001) and the Mass Spectrometer and Incoherent Scatter (MSIS) model (Hedin, 1991; Picone et al., 2002; Emmert et al., 2020).

These models require calculations within specialized programs to generate the needed quantities, so there have been a num-
45 ber of attempts to construct more simple empirical formulas for the conductivity. These are mainly valid for the dayside, where solar extreme ultraviolet (EUV) radiation is the main contribution to ionization. The review paper by Brekke and Moen (1993) lists 12 different formulas spanning the years 1889 to 1992. More recently conductivity formulas were provided by Richmond (1995b), Galand and Richmond (2001), and Wiltberger et al. (2004). Assimilation techniques used in the Kamide-Richmond-Matsushita (KRM) (Kamide et al., 1981) and the Assimilated Mapping of Ionospheric Electrodynamics (AMIE) (Richmond
and Kamide, 1988) methods also need to use conductivities that are derived from such models. As the solar zenith angle is used

in these formulas, they often produce a sharp gradient at the terminator, so Ridley et al. (2004) had added a scattering term to the solar contribution in order to produce a smoother transition over the terminator for a coupled, magnetosphere-ionosphere numerical simulation.

At high latitudes the knowledge of the basic parameters is much less than for the lower latitudes. Due to the auroral ionization and the convection of ionized plasma from the dayside to nightside it is nearly impossible to specify the state of the ionosphere and neutral atmosphere with high accuracy. In fact, the documentation for the IRI model states that "it provides monthly averages in the non-auroral ionosphere for magnetically quiet conditions" (http://ccmc.gsfc.nasa.gov/modelweb/ionos/iri.html). On the night side the ionization due to high-energy auroral particle precipitation contributes most significantly to conductivity enhancements, and this is where there is the greatest uncertainty. As indicated by Liemohn (2020), the space science community needs to reexamine the methods used to calculate the conductivity enhancements due to auroral particle precipitation.

Due to the number of "known unknowns," the ionospheric conductivity in the high-latitude region remains as one of the least-well quantified parameters in geospace and the study of magnetosphere-ionosphere coupling, yet this is where the most important interactions take place. Global numerical models may estimate the conductivities using formulas that include sunlight ionization rates and ionization production rates from precipitating particles, the recombination rates, and the equilibrium densities, and empirical models and various measurements are often used. For example, Fuller-Rowell and Evans (1987) used electron energy influx and energies from National Oceanic and Atmospheric Administration (NOAA) Television Infrared Observation Satellites (TIROS) to build statistical patterns of these data. These were used in physics-based formulas in order to calculate the Pedersen and Hall conductivities as a function of altitude, as well as the hight-integrated values, which were then used to create maps ordered by an auroral activity index.

A similar to method was used by Hardy et al. (1987) using a statistical model of electron flux from Defense Meteorological Satellite Program (DMSP) measurements sorted by the Kp index. They had used empirical formulas derived from computations by Robinson et al. (1987), relating the conductances to the average energy and energy flux of the electrons. Another statistical technique reported by Ahn et al. (1998) had used radar measurements to derive conductivity, and compared these with ground observations of the magnetic perturbations in order to derive empirical relationships between them. They then used measurements of $\Delta B$ to obtain global maps of the conductivity, which were compared with the results by Hardy et al. (1987).

While these statistical models have similar features, they are not in complete agreement. Another issue is that these models use geomagnetic activity indices as the main parameter, and such indices are not readily available in real time. Index-based models do not take the Interplanetary Magnetic Field (IMF) vector into consideration, and it is known that the electric field and field-aligned current (FAC) patterns change significantly as the IMF changes orientation. As a conductivity model is often used with another model or numerical calculation of electric fields and field-aligned currents, unrealistic results are obtained if the boundaries of the models do not properly align, or if they are not based on consistent IMF orientations.

An alternative technique for obtaining the conductivity, named the "the elementary current method," (Amm, 2001) uses multiple satellite and ground magnetometer measurements for deriving the ionospheric currents. This method is based on split-

ting the ionospheric current vector into divergence-free ($\boldsymbol{J}_{df}$) and curl-free ($\boldsymbol{J}_{cf}$) parts. The total height-integrated ionospheric current that is perpendicular to the magnetic field lines is then written as:

$$\boldsymbol{J}_{\perp} = \boldsymbol{J}_{df} + \boldsymbol{J}_{cf} \tag{5}$$

Ground-based magnetometer data are used to derive the "divergence-free" ionospheric currents that are usually associated with Hall currents. The "curl-free" currents are derived from space-based magnetometer measurements that are sensing the field-aligned currents (FAC) that are linked to the divergence of the ionospheric currents. Thus, magnetometer measurements both above the ionosphere and on the ground are required in order to recover the full $\boldsymbol{J}_{\perp}$. More details about the derivations of these currents will follow in a later section.

If there are no neutral winds present, then from Ohm's law for the ionospheric current sheet,

$$\boldsymbol{J}_{\perp} = \Sigma_P \boldsymbol{E}_{\perp} + \Sigma_H \left( \hat{\boldsymbol{B}}_{\perp} \times \boldsymbol{E}_{\perp} \right) \tag{6}$$

if measurements of the electric field is also available then both the Pedersen and Hall conductances can be obtained from:

$$\Sigma_P^* = \frac{\boldsymbol{J}_{\perp} \cdot \boldsymbol{E}_{\perp}}{\left| \boldsymbol{E}_{\perp} \right|^2} \tag{7}$$

$$\Sigma_H^* = \frac{\hat{\boldsymbol{r}} \cdot \left( \boldsymbol{J}_{\perp} \times \boldsymbol{E}_{\perp} \right)}{\left| \boldsymbol{E}_{\perp} \right|^2} \tag{8}$$

As Amm (2001) had stated, "These equations have been derived under the assumption that the magnetic field lines are directed perpendicular to the ionospheric plane. If they are not, the conductance tensor gets off-diagonal elements, and polarization effects have to be included." It was also noted that a small error in the direction of the vectors can produce inaccuracies, especially where the magnitude of the electric field is small. Amm (1998) indicated that the assumption that the magnetic field lines are assumed to be radial does not cause significant errors at latitudes above $45°$.

We have added the * superscripts to the conductivities in Eq. (7) and (8) to indicate that these derivations are approximations, particularly since the effect of the neutral winds are not included, and their influence is assumed to be small (Amm, 1995). As clarified by Amm et al. (2008), "In reality the total effective electric field $\boldsymbol{E}'_{\perp} = \boldsymbol{E}_{\perp} + \boldsymbol{U} \times \boldsymbol{B}$ should be considered, where $\boldsymbol{U}$ is the neutral wind velocity, but the neutral wind velocity is highly height dependent and notoriously difficult to measure; it is also 1–2 orders of magnitude smaller than the plasma velocity in the E-region, and so it is typically set to zero."

In the example presented by Amm (2001) the Spherical Elementary Currents Systems (SECS) (Amm, 1997; Amm and Viljanen, 1999) method is used to obtain the divergence-free currents from "the upward continuation technique for magnetic disturbance fields from the ground to the ionosphere" (Amm and Viljanen, 1999). Magnetometer measurements obtained from sites in Norway, Sweden, and Finland were used in combination with electric field values from the Scandinavian Twin Auroral Radar Experiment (STARE) coherent scatter radar. The method was demonstrated for a small area using simulated magnetic fields above the ionosphere produced by a current vortex, that were compared with measured values from an overhead pass

by the four Cluster II satellites. In another example Amm et al. (2015) use the SECS methods to solve for the electric field, currents, and conductivity in the ionosphere using only measurements from two of the European Space Agency's (ESA) Swarm spacecraft. Solutions were obtained within a region spanning $7°$ in longitude by $20°$ in latitude, that bounded the parallel tracks of the two satellites.

    The notations used in Eq. (6) to (8) closely follow those of Green et al. (2007), who had demonstrated their use to obtain

maps of the height-integrated Pedersen and Hall conductivity over the entire polar region. The horizontal electric field in the ionosphere ($E_\perp$) was obtained from the SuperDARN radar array, with assimilation of drift-meter, electric field measurements on the DMSP satellite along one orbit path. The SuperDARN statistical model (Ruohoniemi and Greenwald, 1996, 2005) was used to help constrain the fit of $E_\perp$. Ground-based magnetometer data were used to construct a map of the divergence-free ionospheric current, $J_{df}$, using a Spherical Cap Harmonic Analysis (SCHA) (Haines, 1988) and the techniques described by

Chapman and Bartels (1940) and Backus (1986). The curl-free current, $J_{cf}$, was derived from magnetic field measurements on the DMSP, Iridium, and Ørsted satellites. All data were gathered over a one-hour period while measurements by the Advanced Composition Explorer (ACE) satellite indicated that the IMF was relatively stable. The final results by Green et al. (2007) showed maps of the derived Pedersen and Hall conductivities in their Figures 9 and 10, with grey regions indicating where there was too much uncertainty in the time-averaged radar measurements. Superposed contours showed the conductances

obtained by combining the Rasmussen et al. (1988) model for the solar EUV contribution with the Hardy et al. (1987) model for the particle precipitation.

    While Green et al. (2007) show results for only one event, Amm's method is perhaps the most direct way to measure the height-integrated ionospheric conductances, and as such it seems reasonable to give the technique a more thorough test. In this paper we use a similar calculation to generate more detailed maps of the conductivity for a wider range of conditions,

including variations in IMF clock angle and dipole tilt angle. Our input data consist of outputs from three separate empirical models that were derived from large data sets: A new model of the electric potentials (Edwards, 2019), a model of the ground-level geomagnetic perturbations (Weimer, 2013), and a new FAC model from satellite magnetometers (Edwards et al., 2020). Due to the need for both electric fields and currents, from magnetic field measurements on both the ground and in space, we prefer to call this the "electrodynamic method."

We emphasize that the height-integrated conductances that we obtain show the relationship between the total, horizontal current and the magnitude of the electric field measured above the ionosphere. Within the ionosphere the horizontal current density varies as a function of altitude, as demonstrated by Mallinckrodt (1985). Due to a finite conductivity parallel to the magnetic field within the ionosphere, the horizontal electric field may also vary with altitude. One minor but important detail that should be mentioned is that traditionally the height-integrated conductance values are obtained by an actual integration of

the conductances that are computed at a range of altitude values using Eq. (1) and (2), as done by Mallinckrodt (1985).

    The coupled magnetosphere-ionosphere system is a complex, dynamic system that is always changing, so often it is questioned whether or not statistical models can provide an accurate representation of the polar electric fields and currents. Nevertheless, all investigations have found consistent and repeatable results that show how the dipole tilt angle and magnitude

and orientation of the IMF control the global-scale configuration of the ionosphere. The Discussion section lists a number of studies done by several groups using different data sets and methods.

## 2 Derivation of the ionospheric electric fields and currents

### 2.1 IMF Measurements

All statistical models used here incorporated measurements of the IMF and solar wind velocity. Complicating the use of these data is the fact that comparisons of measurements of the IMF taken simultaneous with multiple spacecraft in the solar wind revealed unexpected variations in time lags between the detection of similar features (Weimer et al., 2002). Similar variability in the time lags exist between IMF measurements, taken at the Lagrange point named L1, and the arrival of this IMF at the Earth's magnetosphere. We use the technique described by Weimer and King (2008) to reduce the uncertainty in these delays. We propagate the IMF to the Earth's bow shock and then add another fixed delay to account for propagation through the bow shock, the interaction with the magnetosphere, and the time required for the ionosphere to respond and reconfigure.

The paper by Weimer et al. (2010) looked at measuring these time delays using measurements of the ground-level magnetic field, and also discussed the wide range of values for these delays that were reported in previous publications. Correlations with a coupling function of the IMF were computed as a function of the lag time, after prior propagation to the bow shock. Different time periods were used for averaging the IMF values, in multiples of 5 min. Very broad peaks in the correlations were seen, attributed to the variability in the system, with the correlation increasing as the width of the averaging period was increased. The peak correlations were found at lag times of 30 min on the day side and 40 to 45 min on the night side, measured from the start of the 5-min segment at the center. For the purpose of constructing global patterns using spherical harmonics, Weimer et al. (2010) used 25-min averages of the IMF, starting at 45 min before each data sample and ending at 20 min before.

For model development the ionospheric conditions are only associated with solar wind conditions in the past. The initial 20-min lag accounts for the time it takes for the IMF signal to propagate between the bow shock and ionosphere. The use of an averaging window 25-min long (reduced to 20 min for later models) accounts for the time needed for the ionosphere to reconfigure to a new state after changes in the IMF magnitude and orientation.

The calculations using the algorithm by Weimer and King (2008) are time consuming, and need to be done with a database that spans decades. Therefore the IMF data were processed in advance of the model derivations. For greatest efficiency the sliding averages of the solar wind velocity and the Y and Z components of the IMF are stored at 5-min increments for later use.

### 2.2 The electric fields

We use an updated electric potential model by Edwards (2019), which supplements the database from the Dynamics Explorer-2 spacecraft that was used by Weimer (2005b) with a substantially larger number of measurements from the Swarm spacecraft (Lomidze et al., 2019). Previous models had derived the least-error fit of the model coefficients from electric potentials that were obtained by integrating the measured electric field values along the satellite's path during each polar pass. The same,

time-averaged IMF and solar wind values were assigned to all potentials along the entire pass. The latest version of the model is different in that the coefficients are fit directly from the electric field measurements, rather than the integrated potentials. As described in the previous section, the IMF and solar wind values that are associated with each measurement use 20-min averages that were calculated at 5-min intervals, after accounting for all propagation delays.

On all satellites only the component of the electric field in the direction of motion was usable for the model fits. Like the prior version of the model, this latest version is constructed using SCHA (Haines, 1985), with Legendre functions of real, non-integer degree. All model coefficients were calculated using the entire dataset, or parameter space, without prior sorting into bins.

The electric potential model is in a reference frame that is co-rotating with the Earth. Modified magnetic apex coordinates (VanZandt et al., 1972; Richmond, 1995a) are used, and the electric potentials are assumed to be constant along magnetic field lines. The electric fields in the ionosphere are obtained from the derivatives of the potentials that are produced by the model. In all results shown here an altitude of 110 km is used.

## 2.3 The divergence-free currents

The divergence-free currents are obtained from the empirical model of the ground-level magnetic fields by Weimer (2013). This model was constructed from magnetometer measurements at 149 locations during an 8-year time period, along with the simultaneous IMF measurements from the Advanced Composition Explorer (ACE) spacecraft. All sites are located in the Northern hemisphere, extending down to the magnetic equator. Quiet-time, baseline values were subtracted from the measured magnetic fields, as described in detail by Weimer et al. (2010). The processing is similar to that done by the SuperMAG project (Gjerloev, 2012), which did not have any results publicly available when that work had begun.

The magnetic field measurements were translated and rotated to the magnetic apex coordinate system for use in the construction of the model. The model produces values for the Northward, Eastward, and Vertical components of the magnetic field perturbations given a specification of the Y and Z components of the IMF in Geocentric Solar Magnetic (GSM) coordinates, the solar wind velocity, dipole tilt angle, and the $F_{10.7}$ index of solar radiation. The three components were modeled separately, without use of a scalar potential, and implicitly included the effects of internal, image sources.

The formulas described by Chapman and Bartels (1940), Haines (1988), and Haines and Torta (1994) were used to derive the "ionospheric equivalent current function" (Kamide et al., 1981; Richmond and Kamide, 1988). A detailed description of the process is provided by Weimer (2019), which includes the separation of the magnetic effects into their internal and external sources. The magnetic fields are calculated from the gradient of a scalar potential. A SCHA technique is employed, but since the size of the spherical cap is $90°$ the associated Legendre polynomials with integer degree are used, rather than Legendre functions of real, non-integer degrees. The end result is an expression for the external currents in terms of spherical harmonics:

$$\psi_E(\theta, \lambda) = \frac{a}{\mu_o} \sum_{k=1}^{34} \sum_{m=0}^{min(k,3)} \frac{2k+1}{k+1} \left(\frac{R_2}{a}\right)^k P_k^m(\cos\theta)(g_k^{m,e}\cos m\lambda + h_k^{m,e}\sin m\lambda) \tag{9}$$

where $R_2$ is the radius of the spherical shell on which the external currents are assumed to flow, and $a$ is the radius of the Earth. This "equivalent current" function $\psi_E$ has units of Amperes (or kA). The current density vector is obtained from the negative gradient of this function, rotated by $90°$. We use a spherical shell at an altitude of 110 km. External currents in the magnetosphere are also projected to this shell, including the ring current. As shown by Weimer (2019), better results are

215 obtained if adjustments are made to compensate for such current. At low latitudes the Solar Quiet ($Sq^o$) current systems also appear in these results (Matsushita, 1975), along with the magnetic effects of inter-hemispheric, field-aligned currents, and magnetospheric currents (Yamazaki and Maute, 2017).

## 2.4 The curl-free currents

The new FAC model that we use was developed using a very large database of magnetic field measurements from the Ørsted,

Challenging Mini-satellite Payload (CHAMP), and Swarm missions, along with IMF values from ACE (Edwards et al., 2020). The magnetic field instruments on these satellites have better accuracy and sampling rates than the Iridium satellites used for the Active Magnetosphere and Planetary Electrodynamics Response Experiment (AMPERE) (Anderson et al., 2008, 2014).

Like the previous version of the model (Weimer, 2005b), this new FAC model is constructed using SCHA and it is based on the mathematical derivations by Backus (1986), along with Maxwell's equations. The field-aligned currents are related to the

225 the magnetic field perturbations above the ionosphere by

$$\mu_o \boldsymbol{J} = \nabla \times \Delta \boldsymbol{B}_\perp \tag{10}$$

where $\Delta \boldsymbol{B}_\perp$ are the magnetic perturbations in the plane perpendicular to the currents. Following Backus (1986), the radial FAC is a poloidal current that is related to a toroidal magnetic field, such that

$$\mu_o J_\parallel \hat{\boldsymbol{r}} = \nabla \times (\hat{\boldsymbol{r}} \times \nabla_\perp \psi) \tag{11}$$

where $\psi$ is a "toroidal scalar" that has units of length times magnetic induction (Tm, or more commonly used, cTm). $\nabla_\perp$ is a horizontal (perpendicular) surface gradient that Backus (1986) refers to as $\nabla_S$. This last equation reduces to

$$J_\parallel = \nabla_\perp^2 \psi / \mu_o \tag{12}$$

As (12) can also be written as

$$J_\parallel = \nabla_\perp \cdot (\nabla_\perp \psi / \mu_o) = \nabla_\perp \cdot \boldsymbol{J}_{cf} \tag{13}$$

it is seen the FAC density is related to the divergence of the curl-free "potential current," where

$$\boldsymbol{J}_{cf} = \nabla_\perp \psi / \mu_o = -\hat{\boldsymbol{r}} \times \Delta \boldsymbol{B}_\perp / \mu_o \tag{14}$$

and $\hat{\boldsymbol{r}}$ is downward in the direction of the local magnetic field. A positive field-aligned current is also downward. This result indicates that the curl-free current is in the direction of the gradient of the toroidal scalar. Additionally, this gradient is rotated $90°$ from the direction of the toroidal component of the magnetic field, and vice versa.

The newest model by Edwards et al. (2020) differs from the predecessors in that, rather than first integrating the magnetic field measurements to obtain a magnetic potential, the magnetic field values measured on the spacecraft are used directly in the least-error fits. Preprocessing of the data involves subtraction of the Earth's internal field and translation into magnetic apex coordinates. As in the case of the electric field model, the IMF and solar wind values were averaged over a 20-min window, at 5-min increments. The FAC is calculated directly from Eq. (10), rather than Eq. (12), and the curl-free currents are calculated

from the right side of Eq. (14), rather than the middle part. In other words, rotating the modeled magnetic field by $90°$ and dividing by $\mu_o$ provides the curl-free component of $\boldsymbol{J}_\perp$ needed to solve for the conductivity with Eq. (7) and (8).

An earlier work by Edwards et al. (2017) had compared the field-aligned currents with four different indices of solar radiation defined by Tobiska et al. (2008). The expected correlations between these indices and the FAC were verified, so the latest FAC model was constructed so that it could use any one of these indices. Since the ground-level magnetic field model had used only

245 the $F_{10.7}$ index, this same index is the one that is used for the FAC model results that are presented here.

## 3 Poynting flux and Joule heating

In the results that follow we also include comparative maps of the distribution of the perturbation Poynting flux and Joule heating. The perturbation Poynting vector is calculated from the electric field and perturbation magnetic field that is perpendicular to the field-lines carrying the current:

$$\boldsymbol{S}_p = \boldsymbol{E}_\perp \times \Delta \boldsymbol{B}_\perp / \mu_o \tag{15}$$

where $\mu_o$ is the permeability of free space. As this perturbation Poynting vector has just one component that is parallel to the current flow, the magnitude of the vector (the rate of energy flow through a spherical surface) is frequently referred to as simply Poynting flux. While it is possible to calculate a Poynting flux using the full geomagnetic field, in the absence of currents the curl of this field is zero. As indicated by Kelley et al. (1991), a Poynting flux calculated with a curl-free magnetic field has no divergence within a closed surface, and it is essentially useless; currents need to be present for energy to be dissipated within

a closed region. Another important point mentioned by Kelley et al. (1991) is that "the Poynting flux yields the correct energy input even if a neutral wind is present in the ionosphere, which is almost always the case."

As we have available the two-component horizontal current, it is also useful calculate the distribution of the Joule heating:

$$H_J = (\boldsymbol{J}_{df} + \boldsymbol{J}_{cf}) \cdot \boldsymbol{E}_\perp \tag{16}$$

Comparing these two quantities is useful for the simple reason that the perturbation Poynting flux can be obtained from spacecraft measurements, while the distribution of the ionospheric Joule heating cannot be easily measured or calculated,

even though the later is the quantity that is more desired. As pointed out by Richmond (2010), these two quantities are not necessarily the same, and it was postulated that "the associated perturbation Poynting flux can possibly be very different from the integrated energy dissipation below."

## 4 Results

In the figures that follow are shown results of the ionospheric conductivity calculations using Eq. (7) and (8), for different
combinations of dipole tilt angle (Laundal and Richmond, 2017) and the IMF clock angle (the arc tangent of the Y and
Z components of the IMF in GCM coordinates). Also shown are maps of the quantities used to obtain these results, along
with the associated mappings of the perturbation Poynting flux and Joule heating. These additional maps provide useful and
interesting information. All examples are for idealized conditions (not for specific events) that assume steady state solar wind
and IMF values.

Figure 1 shows results with an IMF magnitude of 10 nT in the GSM Y-Z plane input to the models, at a clock angle
orientation of $180°$ (entirely southward, or $B_Z = -10$), and a solar wind velocity of $450 \mathrm{~km\,s^{-1}}$. The dipole tilt angle is
$0°$, and the $F_{10.7}$ index 160 sfu. In the top row of the figure, parts (a)-(c) shows the electric potential and the two horizontal
components of the electric field. The longitudes are marked in Magnetic Local Time (MLT), in magnetic apex coordinates,
with the sun at 12 noon. The gray area on the maps show the region that is outside of the spherical cap that is used in the
SCHA functions in the model. The size of this cap varies with IMF conditions. While the derivatives of the potential are
originally calculated in northward and eastward polar coordinates, it is more useful to convert these to duskward and sunward
components for display and use in the calculations. For example, the typical electric potential pattern has a strong electric field
in the duskward direction, directed from the positive peak on the dawn side toward the negative valley on the dusk side. If the
northward and eastward components are shown then this pattern is not at all obvious. Minimum and maximum values of the
potential and electric fields are indicated in the lower left and right corners of all contour maps, and the locations where these
values are found are marked on the map with the diamond and plus symbols respectively. For clarity the levels chosen for the
counter lines avoid values at exactly zero, as the contouring algorithm tends to entirely miss the zero contour around one of the
two convection cells. As mentioned before, these potentials are in a co-rotating frame of reference.

In the second row of Fig. 1, parts (d)-(f) show the equivalent current function and the duskward and sunward components of
the divergence-free currents that are calculated from the gradients of this function. Since the current flows along the direction
of the contour lines, clockwise around the positive peak, the sunward component of the current flow has some resemblance to
the duskward electric field. As these maps are derived from the magnetic perturbation model that covers the entire hemisphere
(in magnetic apex coordinates), there is no gray boundary. The color bar scale for all horizontal currents is adjusted to approx-
imately match the largest magnitude of the sunward current. Currents outside of the the electric field convection pattern appear
at lower latitudes, having opposite signs. As will be seen in other examples, the patterns that are found at lower latitudes have a
strong dependence on the magnitude and orientation of the IMF. This behavior leads us to assume that these reversed currents
at lower latitudes are due to the magnetic effects of magnetopause and field-aligned currents, that produce a false signature of
ionospheric flow in the equivalent current function.

In the third row of Fig. 1, parts (g)-(i) show the field-aligned current and the duskward and sunward components of the
curl-free current. Via Eq. (14), these currents are just the duskward and sunward magnetic fields, transposed with one sign
change, and divided by $\mu_o$. A predominantly sunward magnetic field in the polar cap translates to a duskward current. These

currents closely resemble the electric fields, as expected. These two components of the magnetic fields were produced directly from the SCHA functions in the FAC model, and then the FAC is calculated from their curl, Eq. (10). This model version (Edwards et al., 2020) had fit the spacecraft magnetic field measurements to the duskward and sunward components in order to reduce the spurious, circular harmonics in the FAC that tend to result when using polar coordinates. The total sums of the upward (negative) and downward (positive) FAC, integrated over the spherical cap, are indicated in the upper left and right corners of the contour map in units of millions of Amperes (MA). As the density of contour lines in the FAC maps tends to get too crowded around the largest values, lines are drawn only for every third interval marked on the color bar. As before, the gray area on the maps show the region outside of the SCHA cap defined by the FAC model.

The fourth row starts with a map of the perturbation Poynting flux in the downward direction, Fig. 1(j), calculated with Eq. (15). The total energy flow into the ionosphere is in the upper-right corner, in Giga-watts (GW). We note that the new electric potential and FAC models produce Poynting flux maps that we consider to be more realistic that the results from the prior models (Weimer, 2005a), with levels that are higher within the polar cap and near the cusp. Sometimes there may be a slight mismatch between the electric potential model and the FAC model (derived from independent data sets), that results in the electric and magnetic fields reversing directions at not exactly the same locations; such misalignment manifests as a negative value of the Poynting flux. These negative fluxes are simply artifacts, and colored in lighter shades of gray. In general the two models match up very well at the electric field reversals, and these areas are rather small in size and magnitude. As with the FAC map, some contour lines are omitted for clarity.

The second map in the fourth row, Fig. 1(k), shows the Pedersen conductivity that is calculated with Eq. (7), but without including the divergence-free current, from the equivalent current function. While this result is not physically meaningful, it is useful to include the map for diagnostic purposes and to show the wrong answer that results if the total current is not used. As the calculation in Eq. (7) doesn't work where the magnitude of the electric field is very low, locations where this magnitude is small are flagged as invalid and colored in gray on the map, in addition to latitudes that are below the spherical cap of the electric field or FAC models. The limiting electric field magnitude is 3 mV/m or 7% of the peak magnitude, whichever is greater. It would be desirable to increase the limiting electric field strength, to around 10 mV/m for example. The outcome with increasing this limit is that a greater number of results are marked as invalid within the auroral oval.

At the electric field reversals there are small areas where conductivity may appear to be negative or have abnormally high values. Negative values are indicated with a blue coloring on the map, but these values are not considered to be realistic or meaningful. Likewise, large, positive values near the convections reversals should be ignored. In all maps of conductivity the maximum values that are indicated in the lower-right corner of the maps excluded results at latitudes greater than $68°$, in order to avoid the areas around the convection reversals. The color bars on all conductivity maps have a fixed range, unlike the others that are adjusted to accommodate the largest values. A green color shows where the conductivity is greater than zero but less than 3 mho.

The next map, Fig. 1(l), shows the Hall conductivity that is calculated with Eq. (8), but without using the curl-free current from the FAC model. The format is the same as the Pedersen conductivity. While this result is incorrect (as was Fig. 1(k)), it is useful to include as it shows why the divergence-free component alone is not sufficient for calculating the Hall conductivity, as

might be assumed. Including these maps is useful for understanding the contributions that both components have on the final numerical results. In this map there are regions where the derived conductivity is negative, which is unrealistic. These areas are marked in shades of blue that darken as the value becomes more negative. While the alignment between the electric potential and equivalent current functions in Fig. 1(a) and 1(d) is generally good, on the dawn side these negative values appear where the current function reverses direction from the clockwise flow around the positive convection cell, or counter-clockwise around the negative convection cell. As we had mentioned earlier, it is thought that the reversed flows, and the unrealistic, negative conductivity values, result from interference from magnetospheric currents. All locations with negative values are considered to have invalid results.

The bottom row in Fig. 1 shows the results using the total currents, with the two components of the current combined together. At the left, Fig. 1(m) shows the Joule heating that is calculated with from the dot product of the electric field and this total current with Eq. (16). The differences between the Joule heating and the perturbation Poynting flux maps will be discussed in more detail in Section 6. Finally, Fig. 1(n) and 1(o) show the derived values of the Pedersen and Hall conductivities, using the total currents. The auroral oval is easily seen in these results, where the conductivity changes to values greater than 3 mho. The Hall conductivity has enhanced values near 6 MLT, that peak at 32 mho, while the largest Pedersen conductivity (45 mho) is found near midnight. The regions of higher conductivities in both maps correspond to upward field-aligned current, the blue regions in Fig. 1(g), including where this FAC passes through the gap between downward current near midnight. This is a common feature in all results. On the dawn side the upward current is the lower-latitude belt, often referred to as "Region 2", while on the dusk side it is the inner-belt, called "Region 1." On the other hand, the Pedersen conductivity that is calculated near midnight seems too large, and most likely not realistic. We will return to this subject later.

## 5 Results from other dipole tilt and IMF clock angles

In Fig. 2 and 3 are shown maps for dipole tilt angles of $-23°$ and $+23°$, corresponding to winter and summer conditions, while the zero tilt in Fig. 1 corresponds to near equinox conditions. As the dipole tilt angle varies every day by about $\pm11°$, due to the offset of the magnetic pole from the rotation axis, there is a broad range of dates when the dipole tilt angle is at the specified values; the reference to seasons does not refer to exact dates, but a generalized time period. The format of these Figures is the same as before. Both the Pedersen and Hall conductivities in Fig. 2(n) and (o), peak at 69 and 51 mho respectively, which are greater than for the equinox conditions. The maps for summer conditions in Fig. 3 show peak Pedersen and Hall conductivities of 56 and 35 mho, that are lower than the winter values yet greater than at equinox. The positive tilt angle in the summer produces enhanced conductivities on the dayside, as expected. The enhanced Hall conductivity seen near 6 MLT in all three graphs, Fig. 2(o) in particular, agrees with the results presented by Green et al. (2007) with their proof of concept demonstration, in their Fig. 10.

One feature to note is that while the electric potentials have similar patterns in Fig. 1–3, the equivalent current function rotates as the dipole tilt angle changes, and exhibits a sharp twist near the pole under winter conditions (negative dipole tilt, Fig. 2). Another noticeable feature is found near midnight, where the region of enhanced conductivities passes through the

region in the electric potential patterns where the negative, dusk potential cell wraps around and under the positive cell. This warp in the electric potential patterns, known as the Harang discontinuity (Gjerloev and Hoffman, 2001; Marghitu et al., 2009), does not appear in the equivalent current functions.

Next we turn our attention to other IMF orientation angles. Figures 4 and 5 show graphs for IMF clock angles of $90°$ and $270°$, corresponding to positive and negative values of the Y component, with $B_Z = 0$. The magnitude of the IMF is 10 nT, and the dipole tilt angle is zero, the same as in Fig. 1. In both cases the conductivities are lower than when the IMF is southward ($180°$ clock angle), with conductivity values being lowest at the $270°$ clock angle. Additionally, the electric potentials, total FAC, and total Poynting flux are also much lower than when the clock angle is $180°$. In Fig. 5 the enhanced Pedersen conductivity previously seen near 0 MLT is noticeably absent. The westward electrojet is also reduced, the region of positive duskward current near 0 MLT in subplots (e) and (h) in all examples. Examples of the results for these two IMF clock angles with negative and positive tilt angles (winter and summer) are included in the supplementary information contained at at https://doi.org/10.5281/zenodo.3985988. This supplement also contains a set of graphs showing the same combinations of IMF clock angle and dipole tilt angle, but with the IMF magnitude reduced from 10 to 5 nT. Similar variations in the conductivities are seen in these other examples, such as the lower values when the Y component is negative ($270°$).

In order to get a better comparison of the effects that the dipole tilt and IMF clock angles have on the conductivity values on the dawn and dusk sides, Fig. 6 shows a graph of the Pedersen conductivity as a function of latitude from $50°$ to $75°$, along a meridional slice at 4 hours MLT. Results were calculated for IMF magnitudes of 1 to 29 nT, at 1 nT intervals, and stacked vertically along the ordinate. The graphs are repeated on a three-by-three grid with dipole tilt angles of $-23°$ (left column), $0°$ (middle), and $+23°$ (right column) and IMF clock angles of $90°$ (top row), $180°$ (middle), and $270°$ (bottom row). Conductivities are indicated with the colors with the scale shown at the bottom of the figure. Gray areas on the graph show where there are no valid results because the location was outside of one of the model's lower-latitude boundary, or the electric field magnitude was too low. The patterns shift to lower latitudes as the magnitude of the IMF increases due to the expansion of the auroral ovals. Blue areas also show invalid results, where the divergence-free current function has a sign opposing the electric field. Fig. 7 shows the Hall conductivities in an identical format. Figures 8 and 9 repeat the graphs for the dusk side at 20 MLT. The 4 and 20 MLT slices mostly avoid the artifacts found at both high and low latitudes at all clock angles.

These graphs show that the conductivities have an asymmetrical response to the clock angle variations. On the dawn side at 4 MLT with a $90°$ clock angle (top rows in Fig. 6 and 7) the conductivity values are generally larger than at $270°$ (bottom rows), while both are exceeded when the IMF is at $180°$ (middle rows). The tilt angle that corresponds to winter conditions (left columns) often produce the largest values. On the dusk side (Fig. 8 and 9) the southward IMF ($180°$, middle rows) again produces the larger conductivity values, but the seasonal (tilt angle) differences are not always as significant. Enhancements near $70°$ latitude (at 10 nT IMF) are produced by the upward, Region 1 currents.

## 6  Comparing the maps of Joule heating and Poynting Flux

As mentioned earlier, Richmond (2010) shows how the distributions of the perturbation Poynting flux and the Joule heat need not be identical. The Poynting vector that enters the ionosphere is not necessarily the same as the height-integrated heating rate below. The paper by Vanhamäki et al. (2012) addresses this topic as well, showing that if only the curl-free component of the current is used to compute the Joule heating, then the mathematical result is identical to the Poynting flux distribution (also shown by Weimer (2005a)). This result is simply due to the way the curl-free component is computed from the perturbation magnetic field and mathematical identities. But when the divergence-free component is added to the Joule heating calculation then there are differences in the distribution of energy dissipation. Vanhamäki et al. (2012) explain that "between these areas the Poynting flux is transported horizontally near the ionospheric plane. . . . Conductance gradients give raise to divergence-free Pedersen currents that modify the spatial distribution (but not the total amount) of Joule heating and to curl-free Hall currents that modify the spatial distribution (but not the total amount) of vertical Poynting flux." In their section 3.1 Vanhamäki et al. (2012) show that the dot product of the horizontal, curl-free electric field with $\boldsymbol{J}_{df}$ should integrate to zero within the boundary where the electric field goes to zero, the result of the mathematical identities, which is why the totals are the same.

Our results in Fig. 1 through 5 do show differences between the maps of the Joule heating (m) and the perturbation Poynting flux (j). The Joule heating in the high-latitude polar cap tends to be lower than the Poynting flux, and greater in the auroral oval, where the conductances are greater, while the integrated sums are nearly equal. In order to more clearly demonstrate the differences between regions, Fig. 10 shows a map of the dot product of the electric field with the $\boldsymbol{J}_{df}$ component alone, for the same case shown in Fig. 1. The transfer of energy from the polar regions to auroral oval is clear. The numbers in the upper left and right corners show the totals of just the areas with negative and positive values, respectively. The 2 GW difference between the totals is only 0.7% of the 303 GW total Poynting flux, and such differences are always less than 1% of the total Poynting flux. As the sums should be equal, the differences are the result of numerical error. The totals in the graphs were derived from 19531 data points at and above $50°$ latitude, forming 38597 triangles on the spherical cap. Each side of the spherical triangles spans an arc length of roughly $0.5°$. The values of each quantity are evaluated at the vertices, and the mean value within each triangle are multiplied with the triangle's area, and summed.

## 7  Discussion

We have applied three, separate empirical models to the formulas proposed by Amm (2001) to calculate the ionospheric Pedersen and Hall conductivities, producing new maps of these conductivities for various conditions. For the most part, the values of the conductivities that are produced seem reasonable. Enhanced conductivities in the auroral ovals are seen, as expected, with values in the range of 3 to 9 mhos under moderate conditions, with some regions having substantial enhancements on top of that. Another result that was entirely expected is that conductivity values increase as the Z component of the IMF becomes more negative. In general, the range of values that we obtain are within reason, in comparison with the results found by Fuller-Rowell and Evans (1987), Hardy et al. (1987) Ahn et al. (1998), and very recently, Carter et al. (2020). At large IMF

magnitudes the conductivity results that are greater than 40 mho seem too extreme, but such values are also found in the model results reported by Mukhopadhyay et al. (27 August 2020).

At positive and negative values of the Y component of the IMF the conductivity results are very different. This is significant, since the existing statistical models of conductivity (see Introduction) do not account for the orientation of the IMF. Some of the conductivity values that we found seem to be greater than what are produced with existing models. Both the Hall and Pedersen conductivities are higher for winter, or negative dipole tilt, conditions, particularly on the dawn side and toward midnight. There are some gaps and artifacts produced where the various models don't exactly line up. The results shown in Fig. 7 are

in agreement with the findings by Ohtani et al. (2019) that "the nightside westward electrojet (WEJ) is more intense when the ionosphere is dark."

The equivalent currents often have patterns at lower latitudes that cause the conductivity that is calculated to have negative values; such results are not realistic and therefore rejected. Magnetospheric and low-latitude, field-aligned currents are the likely source such results. The correction that was employed to compensate for the ring current actually had little effect on

these results, with differences in the maximum values on the order of 2 mho if the correction was either removed or doubled, so that adjustment doesn't seem to be an issue.

One persistent feature in the results is the presence of the extraordinarily large Pedersen conductivities in a narrow band near midnight. A close examination of the map data shows that the peak value occurs in a region of low electric field strength, exactly where the sunward electric field at midnight passes through zero while changing sign from negative to positive, as

latitude decreases. The curl-free, sunward current changes sign also, about one-half of a degree to the north. The duskward electric field is weak, around $5 \, \mathrm{mV \, m^{-1}}$. There is a rather strong ($> 200 \, \mathrm{mA \, m^{-1}}$) divergence-free (equivalent) current here in the duskward direction, part of the westward, electrojet. This Hall current does not change sign with the electric field, which results in the derived Hall conductivity changing signs from positive to negative. This same pattern is found often, and in association with the Harang region where the negative, dusk electric potential cell wraps around the positive cell near

midnight, on the lower-latitude side (Gjerloev and Hoffman, 2001; Marghitu et al., 2009). When the IMF is in the -Y direction (Fig. 5, $270°$ orientation) the extra large Pedersen conductivity is not present, the signature of the Harang discontinuity is weak, and the westward (duskward) currents at midnight are substantially lower.

Shue and Weimer (1994) had proposed that polarization electric fields around areas of enhanced conductivity are responsible for forming the Harang discontinuity; the effect is to block the divergence of Hall currents where there are gradients in the Hall

conductivity. Further evidence of this concept has been provided by Nakamizo and Yoshikawa (2019), and our results are consistent with this hypothesis. Equations (7) and (8) are not accurate where there are conductivity gradients, so the derived conductivity values near midnight, from 22 to 2 MLT, should not be trusted. Likewise, the results near noon are also suspect.

The effects of the neutral winds, which are difficult to measure, are not included in these results. One result of the neutral winds is that the heating is not entirely ohmic, but includes frictional heating from the relative motion of plasma and neutrals

(Vasyliūnas and Song, 2005), and acceleration of the neutrals.

Thayer (1998) indicates that a neutral wind in the direction of the $\boldsymbol{E} \times \boldsymbol{B}$ vector will decrease the Joule heating rate, while a component of the neutral wind in the opposite direction will act to increase it. His observations in one particular time period

indicated that the neutral winds could reduce the local Joule heating rate by over 75% in the upper E region while enhancing the local heating rate by nearly 50% in the lower E region, with an overall decrease of 40% in the height-integrated values. If the electric field is in a steady direction then the neutral winds act to reduce the Joule heating rate, but if the electric fields change directions suddenly then the effect is reversed (Thayer, 1998).

Billett et al. (2018) report that the reduction in the Joule heating due to the neutral winds primarily happens at high magnetic latitudes and in the dusk sector. They report on observations showing a persistent absence of a neutral winds in the dawn circulation cell, and hence a lower reduction. They report that "the percentage contribution of the wind correction to the area-integrated Joule heating rate can vary by ±14% depending on season and geomagnetic activity level," with a greater influence occurring in the hemispheric winter months.

While our results do not include the effects of the neutral winds, they do show how the currents and electric fields are related to each other under typical conditions, which implicitly includes whatever influence the neutral winds may have. The conductivity values obtained from the electrodynamic models could be what numerical modelers actually need in order to compute electric potentials from the field-aligned currents, or vice versa, if the neutral winds are not available. If the neutral winds are significant, then these results are not indicative of the true conductivity in the usual sense. However, the results do establish a relationship between the electric field above the ionosphere, obtained from measurements, and the currents within the ionosphere, also from measurements in space and on the ground. The current-voltage relationship is really what matters.

Both Richmond (2010) and Vanhamäki et al. (2012) had demonstrated differences between maps of the perturbation Poynting flux and the energy dissipation, but with contrived distributions of energy flow and conductivity. The results shown here are the first demonstration of such differences using more realistic conditions.

All three statistical models provide a large-scale representation of the electric and magnetic field variations. These models all use measurements that were taken as a function of time, while the satellites were moving in space, or in the case of the ground-based magnetometers, fixed in position while moving through local time. The original measurements contain fluctuations at various frequencies, with the most rapid fluctuations, lasting a few seconds or so, often having the largest magnitudes. Examples of these fluctuations can be seen in the recent comparison by Lomidze et al. (2019), showing electric field measurements from the Swarm satellites compared with values from the earlier potential model (Weimer, 2005b). The end-result models are global-scale, spatial representations of the fields that do not include such rapid fluctuations. We consider these variations to be mainly temporal, including a mixture of turbulent features that may persist only for a brief period of time, and perhaps moving in space as well.

With the removal of such fluctuations, the global-scale patterns are consistent and repeatable. For example, the electric potential patterns obtained from double-probe measurements on the Dynamics Explorer-2 satellite (Weimer, 1995, 2005a) are very similar to results obtained from ground radar (Ruohoniemi and Greenwald, 1996, 2005; Cousins and Shepherd, 2010), an electron drift instrument on the Cluster spacecraft (Haaland et al., 2007), and ion drift meters on the Defense Meteorological Satellite Program (DMSP) (Papitashvili and Rich, 2002). The field-aligned current maps obtained from different datasets are also similar and repeatable. For examples, compare the results from magnetometers on Dynamics Explorer-2 (Weimer, 2001, 2005a), DMSP (Papitashvili et al., 2002), the Iridium constellation (Anderson et al., 2008), CHAMP and Swarm (Laundal

et al., 2018), and the Ørsted-CHAMP-Swarm combination (Edwards et al., 2020) that we use here. Laundal et al. (2018) also show maps of the equivalent currents that resemble the ones shown here.

One consequence of the temporal variations is that the errors calculated for each model seem large. During the fitting process the difference between the model outputs and the input values (the total square error) is minimized, and also output with the results. The standard deviations that are calculated are on the order of 20% to 50% of the model outputs at their peak values. Due to the influence of the high-frequency fluctuations, we do not consider such errors to be a true representation of the accuracy of the spatial models, considering the repeatable results just mentioned. As a result, the accuracy of the final conductivity values

are difficult to ascertain with reasonable certainty. Of course, our smoothed calculations of the conductivities do not include the small-scale and meso-scale enhancements that occur within the auroral arcs that are non-stationary in time and space.

A significant source of error is actually in the IMF measurements, since the values that are measured by a spacecraft such as ACE may differ from what actually impacts the Earth's magnetosphere (Weimer et al., 2002; Borovsky, 2018). As indicated by Burkholder et al. (2020), "the solar wind measured by a single spacecraft at L1 often does not impact Earth in a homogeneous

manner." Such uncertainties in the propagated IMF were greater in the electric potential model as it had required use of some IMF measurements from the International Sun-Earth Explorer 3 (ISEE3), which was located farther off to the side in the solar wind. The errors in the propagated IMF measurements cannot be fully evaluated until additional spacecraft are placed in the upstream solar wind. The presence of multiple, random and uncontrolled variables often makes space physics research more difficult than laboratory experiments under controlled conditions.

Mukhopadhyay et al. (27 August 2020) had tested the usefulness of four conductivity models in a numerical simulation by evaluating whether or not there was an improvement in various metrics and skill scores for the predictions of ground-level magnetic perturbations. A similar approach could be used for evaluating the conductivity values derived with the electrodynamic method, including use in ionospheric assimilation codes and coupled, ionosphere-thermosphere models.

## 8 Conclusions

The lack of definitive values for the ionospheric conductivities is a significant problem in magnetospheric physics. Existing empirical models are not entirely in agreement, and typically use indices of activity level rather than solar wind and IMF measurements. Amm (2001) had presented formulas for obtaining the conductivity from measurements of the electric fields, and currents derived indirectly from magnetic field measurements on the ground and in space. The proof of concept demonstration by Green et al. (2007) had shown promising results, which presented an opportunity to try the technique with a combination

of empirical models to obtain estimates of the height-integrated Pedersen and Hall conductivities. Maps of the high-latitude, ionospheric conductivities were derived for varying combinations of the dipole tilt angle, IMF magnitude, and IMF orientation angle in the GSM Y-Z plane. There are places where the technique fails to produce valid conductivity values, so the results are not entirely satisfactory. Nevertheless, these findings should still be of some use to the space science community.

The conductivity maps that were obtained have features that were expected on the basis of prior knowledge, such as en-

530 hancements within the auroral oval and increasing conductivity as the Z component of the IMF becomes more negative. The

dawn and dusk sides do not have a symmetric response with respect to flipping the sign of $B_Y$. It was found that reversing the sign of the Y component of the IMF results in substantially different conductivity patterns, with values that are generally lower when $B_Y$ is negative, corresponding to clock angles near $270°$. Changes in the dipole tilt angle also have a significant influence. These factors need to be considered in the future development of conductivity models and their use.

*Code and data availability.* An archive of graphs and data can be found at https://doi.org/10.5281/zenodo.3985988. This archive includes the supplement with additional illustrations mentioned in the text, and reproductions of Fig. 1 to 5, but arranged in a horizontal, landscape format for easier viewing on a computer screen, and similar figures generated for other cases. Maps of the dot product of the electric field and divergence-free current in each case are also included as well as the equivalent current functions down to a latitude of $35°$. Digital data files containing the conductivity results are contained in the archive, along with Interactive Data Language (IDL) software code needed to
read and interpolate these data. All data needed to generate Fig. 6 to 9 are included, along with the associated graphs, plus additional cases at other IMF clock angles. Altogether, 696 sets are included. The archive contains the ground-level magnetic field model output values that are used to calculate the equivalent currents, the resulting spherical harmonic coefficients, and the SCHA coefficients from the electric potential and FAC models.

Data used in the development of the magnetic perturbation model are listed in the publications by Weimer et al. (2010) and Weimer (2013).
Data used in the development of the field-aligned current model are available through Edwards et al. (2020). The electric potential model (Edwards, 2019) was developed with the Swarm cross-track ion drift data available at http://earth.esa.int/swarm/ and Dynamics Explorer-2 Vector Electric Field measurements at https://cdaweb.gsfc.nasa.gov/. All models used solar wind and IMF measurements from the IMP8, ISEE3, and ACE satellites, at https://cdaweb.gsfc.nasa.gov/.

*Author contributions.* The writing of this article was led by DW with contributions from TE. DW created all figures. TE created the empirical
FAC and electric potential models, and DW created the magnetic perturbation model and the equivalent currents.

*Competing interests.* The authors declare that they have no competing interests.

*Acknowledgements.* Daniel Weimer and Thom Edwards were supported by NSF grants AGS-1638270 and PLR-1543364 to Virginia Tech, with additional support from a subcontract to Hampton University, on NASA grant NNX15AE05G. The authors thank Art Richmond for helpful comments on an earlier version of the manuscript.

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

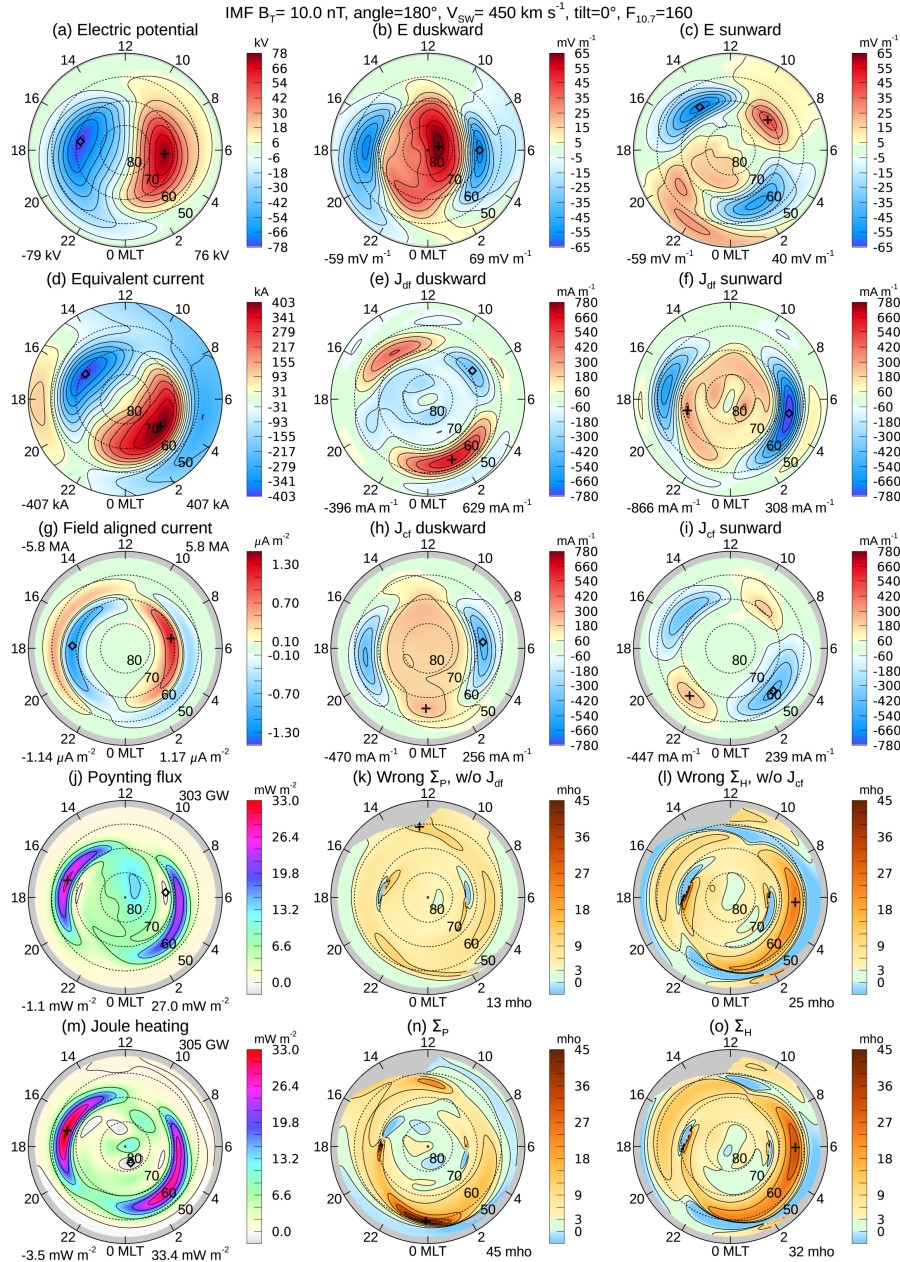

IMF $B_T$= 10.0 nT, angle=180°, $V_{SW}$= 450 km s$^{-1}$, tilt=0°, $F_{10.7}$=160

**Figure 1.** Conductivity input data and results, for IMF $B_T$ magnitude 10 nT at $180°$ clock angle, the solar wind velocity is 450 km s$^{-1}$, the $F_{10.7}$ index is 160 sfu, and the dipole tilt angle is $0°$ corresponding to near equinox. Details are explained in the text.

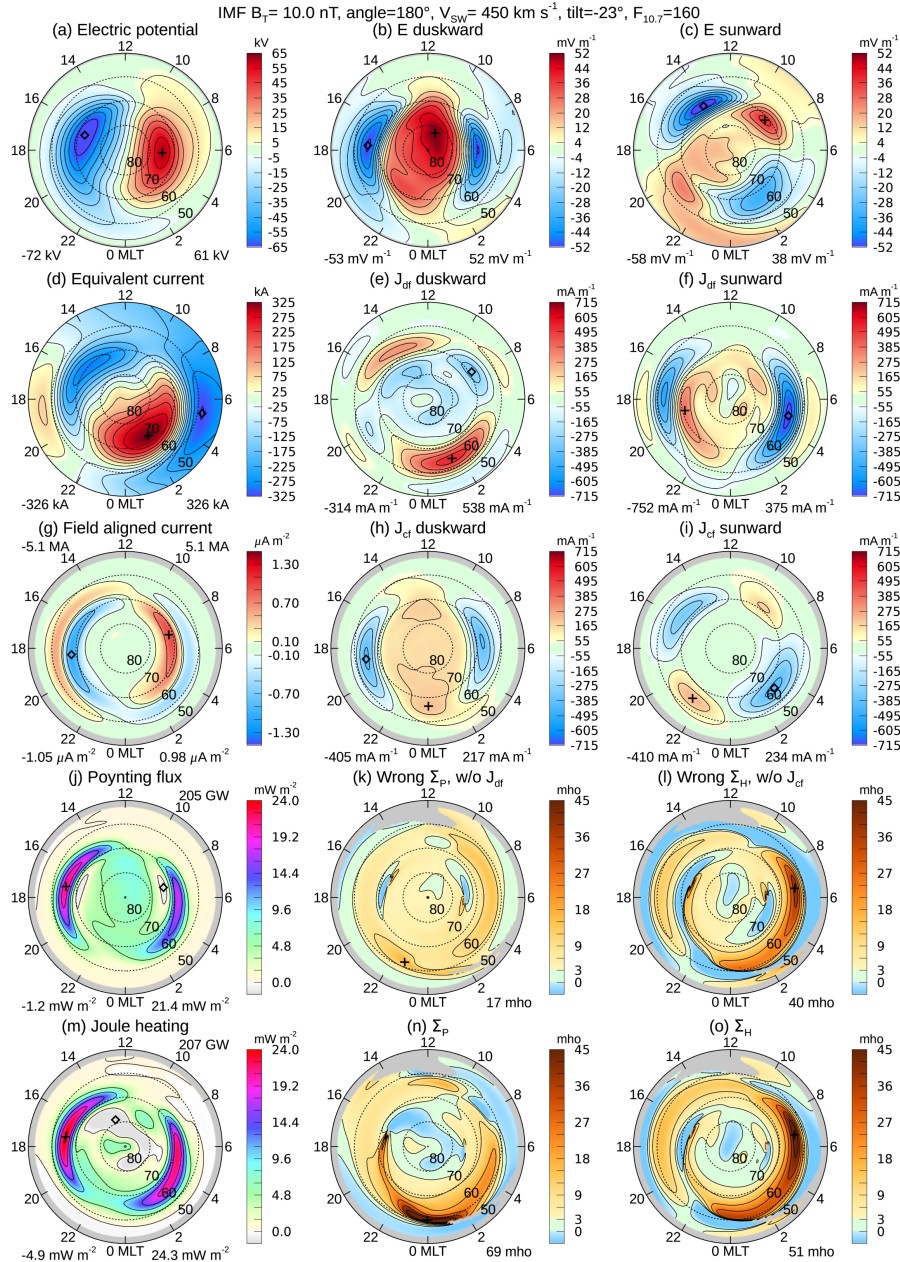

**Figure 2.** Conductivity input data and results for the same conditions as in Fig. 1, except that the dipole tilt angle is for winter conditions. The IMF $B_T$ magnitude is 10 nT at $180°$, the solar wind velocity is $450 \ \text{km s}^{-1}$, the $F_{10.7}$ index 160 sfu, and the dipole tilt angle is $-23°$.

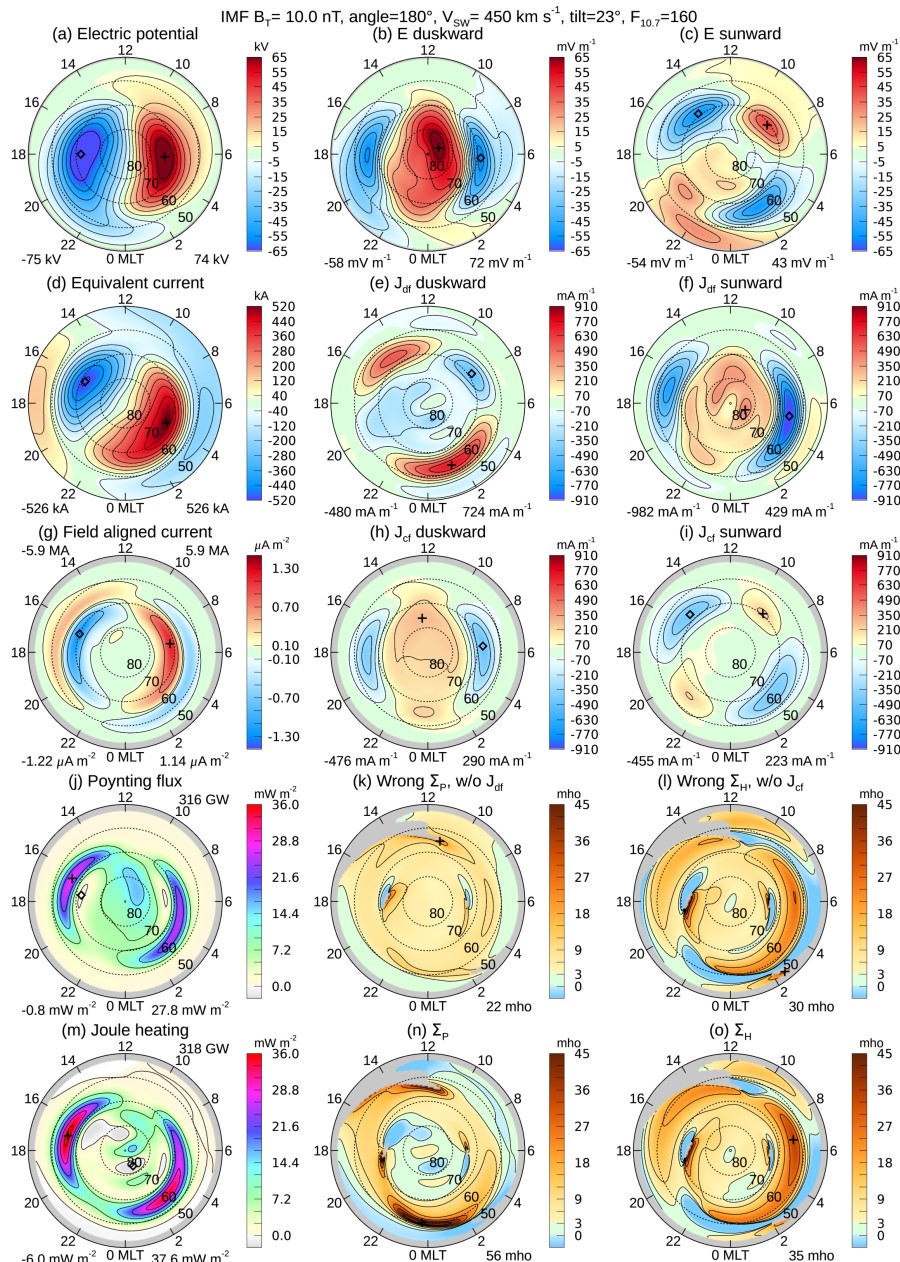

**Figure 3.** Conductivity input data and results for the same conditions as in Fig. 1, except that the dipole tilt angle is for summer conditions. The IMF $B_T$ magnitude is 10 nT at $180°$, the solar wind velocity is $450\,\mathrm{km\,s^{-1}}$, the $F_{10.7}$ index 160 sfu, and the dipole tilt angle is $+23°$.

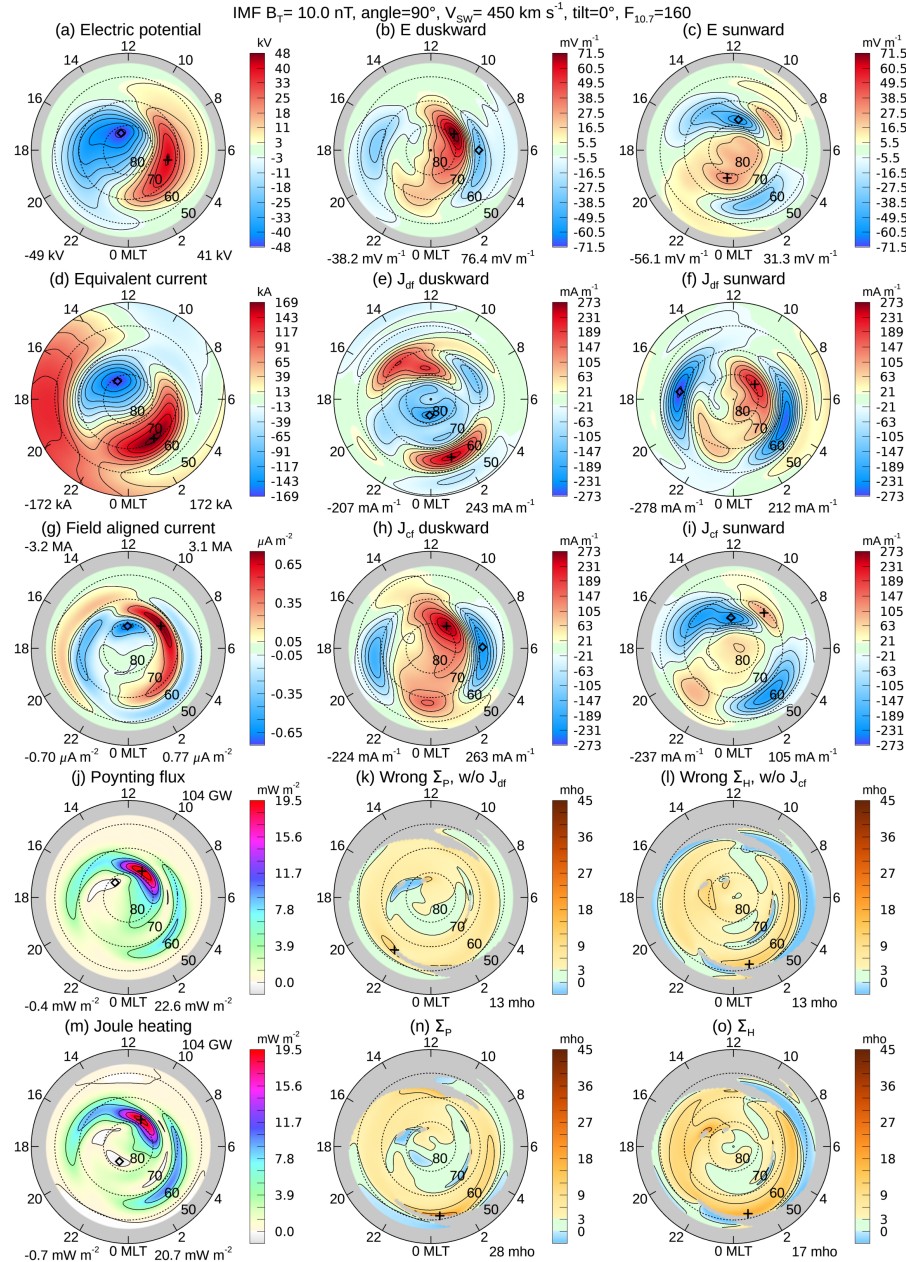

**Figure 4.** Conductivity input data and results for the same conditions as in Fig. 1, except that IMF clock angle is changed $90°$. The IMF $B_T$ magnitude is 10 nT, the solar wind velocity $450 \, \mathrm{km \, s^{-1}}$, the $F_{10.7}$ index 160 sfu, and the dipole tilt angle is $0°$.

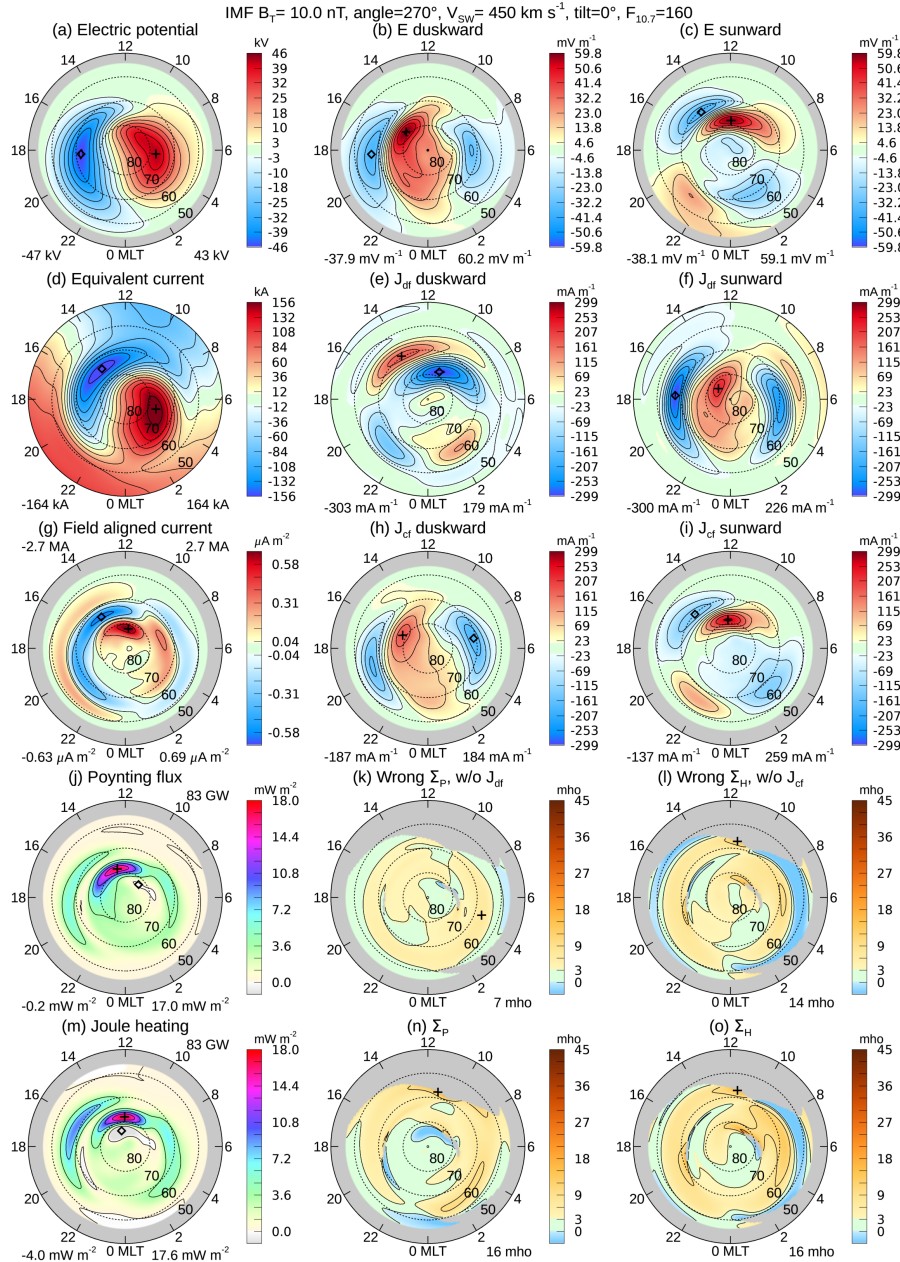

**Figure 5.** Conductivity input data and results for the same conditions as in Fig. 1, except that IMF clock angle is changed $270°$. The IMF $B_T$ magnitude is 10 nT, the solar wind velocity $450\ \mathrm{km\ s^{-1}}$, the $F_{10.7}$ index 160 sfu, and the dipole tilt angle is $0°$.

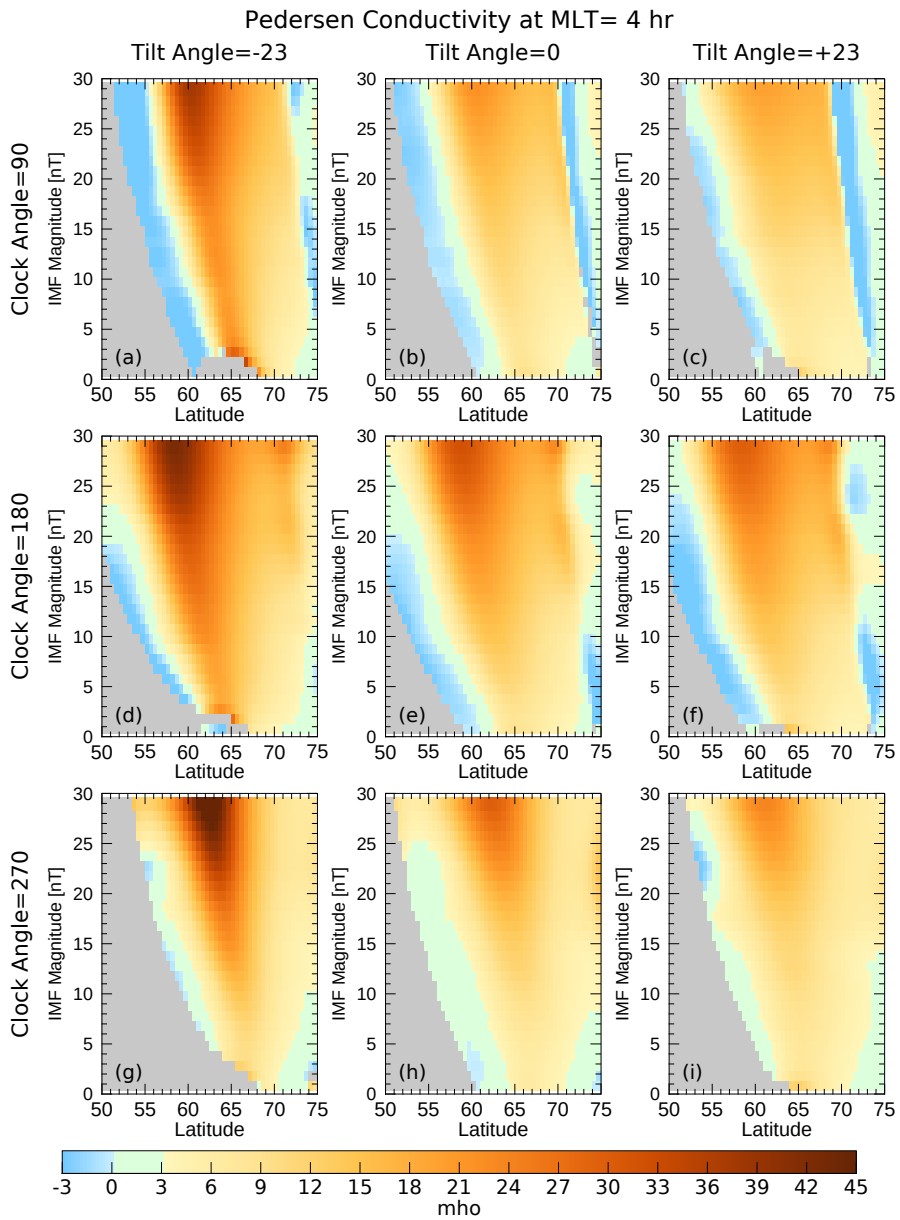

**Figure 6.** Pedersen conductivity as a function of latitude from $50°$ to $75°$, at 4 hours MLT. Values were calculated for IMF magnitudes of 1 to 29 nT, at 1 nT intervals, indicated on the ordinate. Multiple graphs are shown for dipole tilt angles of $-23°$ (left column), $0°$ (middle), and $+23°$ (right column) and IMF clock angles of $90°$ (top row), $180°$ (middle), and $270°$ (bottom row). Conductivity values are colored according to the scale at the bottom. Gray and blue areas indicate invalid results, as noted in the text.

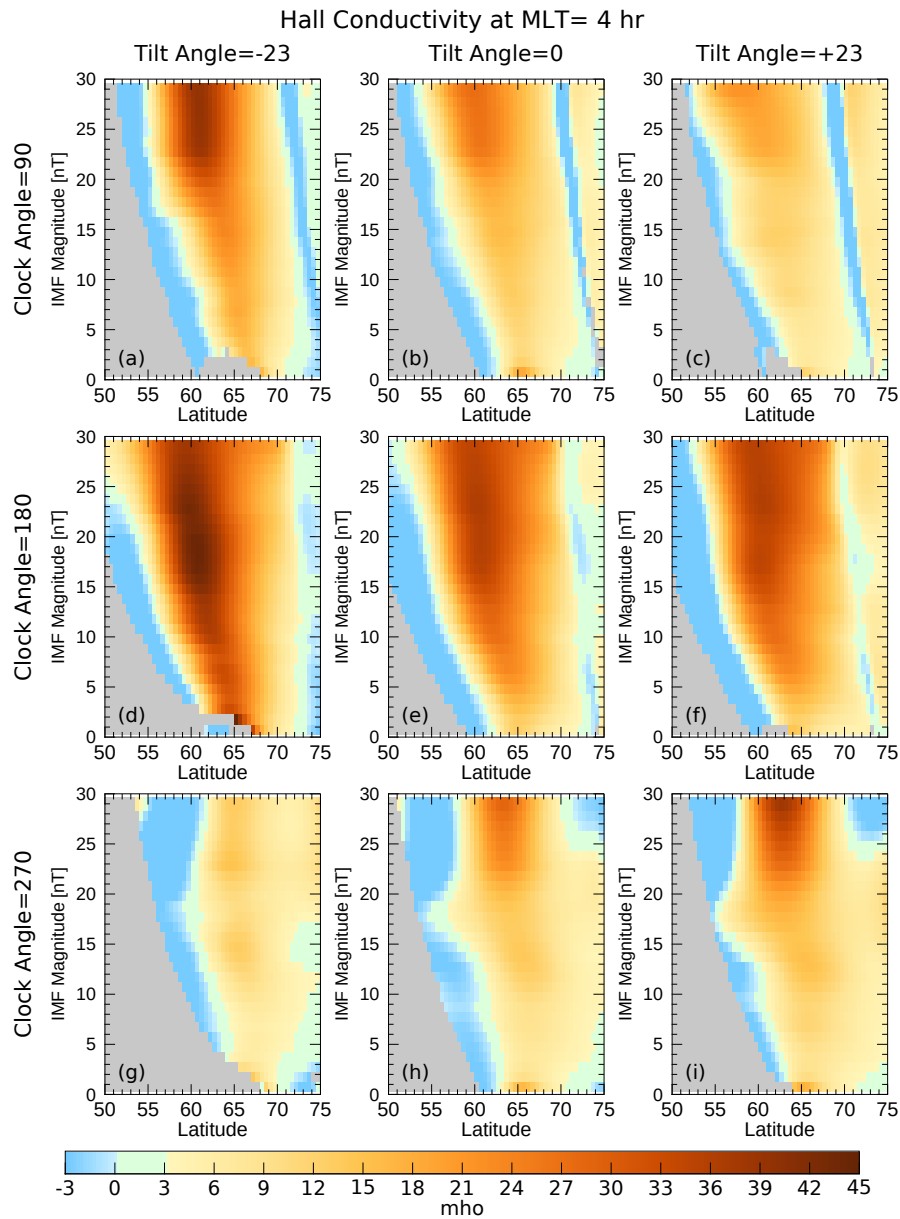

**Figure 7.** Hall conductivity as a function of latitude from $50°$ to $75°$, at 4 hours MLT. The format of the graph is the same as in Fig. 6.

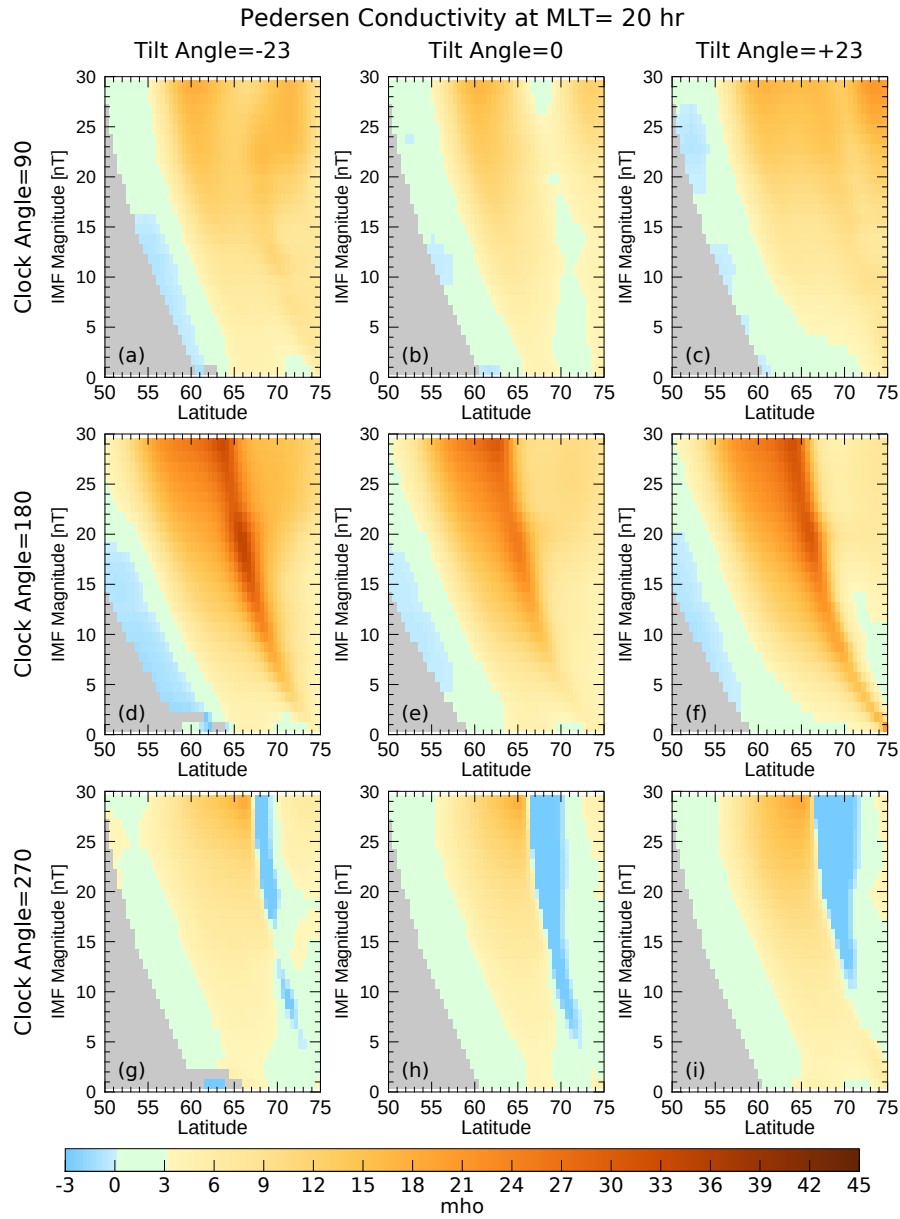

**Figure 8.** Pedersen conductivity as a function of latitude from $50°$ to $75°$, at 20 hours MLT. The format of the graph is the same as in Fig. 6, with only a change in the MLT.

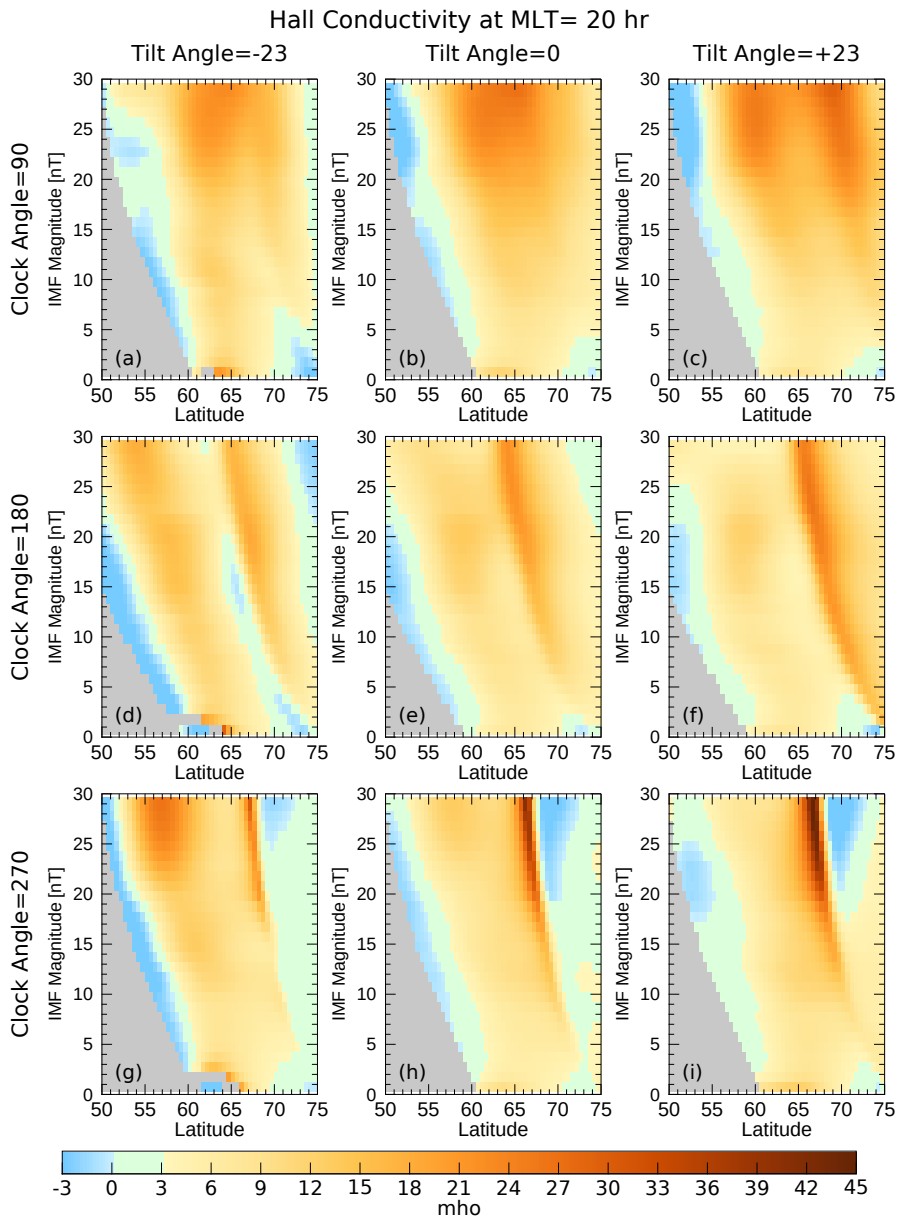

**Figure 9.** Hall conductivity as a function of latitude from $50°$ to $75°$, at 20 hours MLT. The format of the graph is the same as in Fig. 6 to 8.

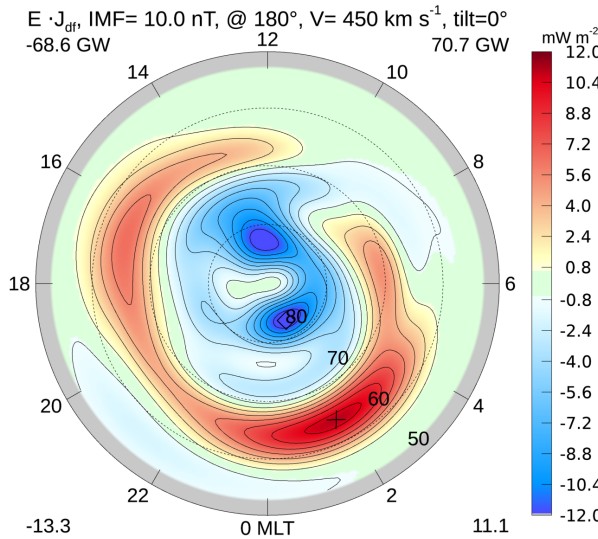

**Figure 10.** Dot product of electric field and divergence-free current, for the same conditions as in Fig. 1. The IMF $B_T$ magnitude is 10 nT at $180°$, the solar wind velocity is $450\,\mathrm{km\,s^{-1}}$, the $F_{10.7}$ index 160 sfu, and the dipole tilt angle is $0°$. The integrated sum of all negative values is indicated in the upper left corner, and this total for all positive values in indicated in the upper right corner, in units of GigaWatts.