# Peer review of "Testing the Electrodynamic Method to Derive Height-Integrated Ionospheric Conductances"

_Annales Geophysicae, 2020_

## Referee Comment (RC1) · Anonymous Referee #1 · 22 Sep 2020

The paper presents a model of height integrated ionospheric conductance that is derived from results of other dedicated empirical models. The manuscript is well written, in most part clear and understandable and potentially suitable for publication at Annales Geophysicae.

There are few comments, that the authors shall addresses:

1. The model is a composite of different empirical models. Quantify the uncertainty and/or error that accumulates by using the empirical models to build another one.

2. This point relates to point 1. The Discussion mentioned a row of error that might be in the model results. Certainly, the small-scale variation cannot be captured, but this is

not what would be expected from the empirical model and is quite clear. Also, as the major source of error, an uncertain solar wind measurement is mentioned. However, it needs to be checked how this is valid and that uncertainties from point 1, may not be more important.

3. The conclusion repeats earlier finding from other authors and mention the shortcomings of the model results submitted here, but that the results might be of some value to the community. This again calls for a quantification, if rough, of the errors expected. To which application the value is expected?

4. L 23 says that earlier formulars are confusing and that the authors applied a simpler formula. What does justify the simplification?

5. I had sometimes the impression that not the most recent developments of certain areas are referred to, such as for AMIE, SECS method, or NRLMSIS. Please check about newer developments in these fields.

6. What is the opinion of the authors if direct observations near the E region would enhance their findings, such as is provided by a satellite mission like Daedalus? (https://daedalus.earth)

---

## Referee Comment (RC2) · Anonymous Referee #2 · 7 Oct 2020

Daniel Weimer and Thom Edwards, Testing the Electrodynamic Method to Derive Height-Integrated Ionospheric Conductances, MS No.: angeo-2020-60

First and foremost I apologize to the authors and editor for the slow reply. I simply got swamped with work.

The paper applies three, separate empirical models to the formulas proposed by Amm (2001) to calculate the distributions of the ionospheric Pedersen and Hall conductivities for various conditions.

The paper is, with some exceptions, well written, figures are of publication quality, the technique is well explained and the paper is logically organized. I do, however, suggest

the authors address my major weaknesses. In particular the methodology needs to be critically discussed. Please view my many comments and concerns as an attempt to be helpful. Based on this is recommend publication after minor revisions.

Major concerns.

Methodology 1. While the technical approach is largely (see below) well explained the methodology is questionable. When the authors organize their data by IMF, dipole tilt angle, F10.7 and so on the assumption is that these parameters are indeed controlling the ionospheric electrodynamics. There is a causality implied in the approach which opens two fundamental questions: is there a cause-and-effect and if so what is the delay?. I think this may work on the dayside where delays between SW conditions and the response of the ionospheric electrodynamics are likely on the order of minutes and thus within the 5-min binning. I, however, object to the approach on the nightside. Here the plasma-sheet is involved and the FACs, particle precipitation, convection and thus conductances and ionospheric currents are due to processes within the plasma sheet. These are only probabilistic related to the SW conditions and the delays are largely unknown and certainly much larger than the 5-min binning. An example is in L338 "IMF clock angles influences the conductivity values". I do not think the word 'influences' is right. Another example is in L203. Is the previous 20 used or is it the center of the window? If centered the ionospheric conditions can hardly be associated with future SW conditions? A discussion of this methodology and arguments for why this is appropriate is required. 1) Methodology 2. I worry that the paper uses a circular argument. The SuperDARN convection distributions use a fill-in model which is IMF driven. Thus, the convection has embedded an IMF dependence which will obviously will be seen in the derived distributions. 2) Technique 1. Please provide an argument for why AMPERE and SuperMAG was not used. These have been carefully validated and provide more comprehensive datasets than those used here. 3) Technique 2. I worry that the ground level perturbations are assumed to be due to an overhead current. This is not the case in the polar cap and equatorward of the auroral oval.

[Figure]

Please comment/discuss. Also see L254-255. 4) Technique 3. Clearly the solutions are problematic in regions of low E but I also worry about regions in which the direction of E varies sharply. Maintaining the relative alignment of the various parameters in such regions is very difficult and mis-alignments result in erroneous conductances since these essentially are fudge-factors. 5) Technique 4. The authors allude in L435 to the Hall vs DF. Can you include a brief discussion on this widely used assumption? Also see L85-86. 6) Energy transfer (e.g. L371). How is energy transported over these long distances? L359 states: "their gradients act to transport energy flux" but I don't understand how this actually works. 7) L437-455. Please change from 'scale sizes' to something like 'small- and meso-scale'. Also, I don't know where the 20%-50% comes from. The contribution from these processes is very much a topic of discussion.

Minor concerns.

Remove L75-77. I disagree, considering all the other complications. L150. "after accounting for solar wind propagation delays". Please elaborate. Which technique is used and I assume this is from L1. L293. "a wrong result again, but useful to include." Can you please add an argument "a wrong result again, but useful to include, because. . ." L319. The Green el al paper was a proof of concept or technical paper. L339: See Ohtani, S., Gjerloev, J. W., Johnsen, M. G., Yamauchi, M., Brändström, U., & Lewis, A. M. (2019). Solar illumination dependence of the auroral electrojet intensity: Interplay between the solar zenith angle and dipole tilt. Journal of Geophysical Research: Space Physics, 124, 6636– 6653. https://doi.org/10.1029/2019JA026707.

Clean up language. For example: "atmosphere with a high level of precision" what does this mean? "that cause the derived conductivity to seem negative", why seem? "gradients act to transport energy flux from", transport flux? "and potentially useful conductivity values are lost." to who and how?

---

## Author Comment (AC1) · 20 Oct 2020

Referee #2:

Methodology 1. While the technical approach is largely (see below) well explained the methodology is questionable. When the authors organize their data by IMF, dipole tilt angle, F10.7 and so on the assumption is that these parameters are indeed controlling the ionospheric electrodynamics. There is a causality implied in the approach which opens two fundamental questions: is there a cause-and-effect and if so what is the delay?. I think this may work on the dayside where delays between SW conditions and the response of the ionospheric electrodynamics are likely on the order of minutes

and thus within the 5-min binning. I, however, object to the approach on the nightside. Here the plasma-sheet is involved and the FACs, particle precipitation, convection and thus conductances and ionospheric currents are due to processes within the plasma sheet. These are only probabilistic related to the SW conditions and the delays are largely unknown and certainly much larger than the 5-min binning. An example is in L338 "IMF clock angles influences the conductivity values". I do not think the word 'influences' is right. Another example is in L203. Is the previous 20 used or is it the center of the window? If centered the ionospheric conditions can hardly be associated with future SW conditions? A discussion of this methodology and arguments for why this is appropriate is required.

________________-

Response:

This reviewer is correct that the dayside and nightside have different response times after the IMF changes, and that there are uncertainties in the delays. Apparently a revision to the paper needs to be made to clarify that the IMF parameters are averaged over a 20-minute time window, not 5 minutes. This averaging accounts for the fact that the magnetosphere and ionosphere as a whole system requires time to reconfigure after a change in the IMF, and that the nightside takes longer than the dayside. An entire section and several figures in the Weimer et al. [2010] paper is devoted to this topic of variable delays and reconfiguration times, as well as a discussion of the various and sometimes conflicting numbers reported in other papers. Complicating the subject is the fact that there are variable time delays between the measurement of the IMF at the L1 orbit and the time it reaches the Earth [Weimer et al, 2002]. Uncertainty in these delays are reduced by using the technique reported by Weimer and King [2008], to propagate the IMF to the Earth's bow shock. Weimer et al. [2010] had reported on the results of correlations between the ground-level magnetic field response and a function of the IMF (after applying this propagation). Due to the variability in the system it was found that the dayside response was slightly shorter than the nightside

response, but the peaks in the correlation vs. delay curves were very broad. For the purpose of constructing global patterns using spherical harmonics, it was decided that the best compromise was to use a 25-min average of the IMF, after including a 20-min time lag from the computed time of impact at the bow shock. In later papers the averaging period was reduced to 20 minutes. The computation is done so that each measurement (such as magnetic or electric field) is compared with the average of the IMF that had interacted with the magnetosphere over the previous 20-min time period, not a time-centered average. Future solar wind conditions are never used. In order to make the size of the processed IMF database more manageable, the sliding averages were sampled at 5-min intervals, rather than something like 16 seconds. These time-taged IMF values are assigned to the data samples taken at the times falling within each window, for use in the fitting process.

On the night side the processes mentioned by the reviewer add even more uncertainty, particularly due to the irregular and unpredictable occurence of substorms. Nevertheless, in all previous modeling work, including that done by several other authors with different data sets, such as SuperDARN, DMSP, AMPERE, and Cluster) it has been found that the ionospheric electric fields and currents have consistent and repeatable patterns in response to the magnitude and orientation of the IMF and dipole tilt angle. Additional references about these consistent patterns could be added to the revision. The 2017 study by Edwards et al. had found that the solar indices do have a correlations with the magnitude of the field-aligned currents, so it was logical to include the F10.7 index that was used with the ground-level magnetic perturbation model. As the existing empirical models of ionospheric conductivity rely only on activity indices while ignoring the IMF, it seemed that this study needed to be done.

In the revision we will work to find an alternative wording for "IMF clock angles influences the conductivity values". We will also add additional references on the topic of propagation delays:

Weimer, D. R., D. M. Ober, N. C. Maynard, W. J. Burke, M. R. Collier, D. J. McComas,

T. Nagai, and C. W. Smith. Variable time delays in the propagation of the interplanetary magnetic field. J. Geophys. Res., 107((A8)), 2002.

Weimer, D. R., and J. H. King. Improved calculations of interplanetary magnetic field phase front angles and propagation time delays. J. Geophys. Res., 113, 2008.

————————————

Referee #2:

1) Methodology 2. I worry that the paper uses a circular argument. The SuperDARN convection distributions use a fill-in model which is IMF driven. Thus, the convection has embedded an IMF dependence which will obviously will be seen in the derived distributions.

————————————

Response:

"SuperDARN" was mentioned only in the context of the earlier results obtained by Green et al [2007]. This study used an electric potential model derived from the Dynamics Explorer-2 and Swarm satellites, (no SuperDARN data are used) along with the IMF measurements. As the intent is to determine how the IMF influences the resulting conductivity mappings, we do not understand the comment about a circular argument.

————————————-

Referee #2:

2) Technique 1. Please provide an argument for why AMPERE and SuperMAG was not used. These have been carefully validated and provide more comprehensive datasets than those used here.

————————————

Response:

The publications by Weimer et al. [2010] and Weimer [2013] were the results of a project that had started in late 2008, before there were any data available from the SuperMAG web site. So the PI had no other choice but to independently collect and process the magnetometer data from multiple sources, as described in the 2010 paper. After the 2013 magnetic perturbation model was completed there was never any reason or funding to do the same thing all over again with data from SuperMAG. The way that the SuperMAG data are obtained through their web site also makes it nearly impossible to obtain all of the highest resolution data from every Northern hemisphere magnetometer over an eight-year period.

We disagree that the AMPERE dataset is better for this purpose. The magnetometers on the Iridium satellites actually have a very poor resolution and low sampling rate, and their attitude pointing accuracy isn't that great either. The Oersted, CHAMP, and Swarm satellites all have much better magnetometers and attitude control. AMPERE is better suited for event studies, rather than building statistical models, although there are FAC models that have been created from the AMPERE data that strongly resemble ours. The Edwards et al [2020] paper actually includes a comparison of the new FAC model with some AMPERE results, along with a different FAC model.

————————-

Referee #2:

3) Technique 2. I worry that the ground level perturbations are assumed to be due to an overhead current. This is not the case in the polar cap and equatorward of the auroral oval. Please comment/discuss. Also see L254-255.

——————————

Response:

The formulas used to obtain the ionospheric equivalent current functions, described by Chapman and Bartels (1940), Haines (1988), and Haines and Torta (1994) do not

assume that the magnetic perturbations are due only to currents that are directly over-head. The effect of all currents are included in the current function at each point, unlike other methods that simply rotate magnetometer measurements by 90 degrees and interpolate. As mentioned in the Weimer [2019] paper, the method also has the unde-sired side effect of including the effects of magnetospheric and field-aligned currents, which is what the text in L254-255 refers to. The magnetic fields from these external currents produces unrealistic equivalent currents at very low latitudes, often in a direc-tion that is opposite to what is expected from the electric potential model. As indicated in the paper, the conductivity cannot be calculated in these regions.

—————————-

Referee #2:

4) Technique 3. Clearly the solutions are problematic in regions of low E but I also worry about regions in which the direction of E varies sharply. Maintaining the relative alignment of the various parameters in such regions is very difficult and mis-alignments result in erroneous conductances since these essentially are fudge-factors.

—————————

Response:

No fudge-factors are used in any attempt to align the models, if that is what the referee meant. The original model outputs are used as is. As the results are flagged as invalid in the areas where the electric field is low and where the conductivity is calculated as negative, there shouldn't be a problem.

—————————-

Referee #2:

5) Technique 4. The authors allude in L435 to the Hall vs DF. Can you include a brief discussion on this widely used assumption? Also see L85-86.

———————————

Response:

We had comments from a theoretician on a prior version of the paper that the use of the Jdf and Jcf current systems is only a mathematical device, so this sentence was added as an attempt to respond to this remark. But since there is no reference that can be quoted here, we think it is best to just remove this comment in the revision. As the referenced paper by Vanhamäki et al. [2012], and others, have detailed discussions of the Jdf and Jcf components, we don't think that there is much that can be added.

———————————-

Referee #2

6) Energy transfer (e.g. L371). How is energy transported over these long distances? L359 states: "their gradients act to transport energy flux" but I don't understand how this actually works.

———————————

Response:

Line 359 is referring to results in the paper by Vanhamäki et al. [2012]. The energy transfer actually takes place through a horizontal Poynting flux. Paragraph 23 in this paper says "Between these areas the Poynting flux is transported horizontally near the ionospheric plane" and Paragraph 43 says "conductance gradients give raise to divergence-free Pedersen currents that modify the spatial distribution (but not the total amount) of Joule heating and to curl-free Hall currents that modify the spatial distribution (but not the total amount) of vertical Poynting flux."

———————————-

Referee #2:

7) L437-455. Please change from 'scale sizes' to something like 'small- and meso-scale'. Also, I don't know where the 20%-50% comes from. The contribution from these processes is very much a topic of discussion.

———————————

Response:

The text already uses "small-scale," and we can't find any occurence of 'scale sizes'.

The empirical models all use measurements that were taken as a function of time, while the satellites were moving in space, or in the case of the ground-based magnetometers, fixed in position while moving through local time. The original measurements contain fluctuations at various frequencies, with the most rapid fluctuations, lasting a few seconds or so, often having the largest magnitudes. The end-result models are global-scale, spatial representations of the fields that do not have the rapid fluctuations included. During the fitting process the difference between the models and input values (the total square error) is minimized, and this error can be calculated. This is where the 20%-50% numbers come from. But since the high-frequency fluctuations are believed to be mostly temporal variations, rather than spatial, the least-square error that is calculated is not a true representation of the accuracy of the spatial models.

The referee comments indicate that we need to explain this part better in the revision.

—————————————-

Referee #2:

Remove L75-77. I disagree, considering all the other complications.

———————————

Response:

These lines will be rewritten to change the wording of this phrase, which in retrospect is

too harsh: "A larger problem is that models based on activity indices have only marginal utility." The point we were trying to get across is that the community is moving away from "index based" model input and toward real data for a reason, particularly because indices have long lag times before they become available. The need for conductivity mappings that are based on incoming IMF/solar wind values, rather than activity indices, is a major focus point of this paper, and our results show the orientation of the IMF is an important parameter.

________________-

Referee #2:

L150. "after accounting for solar wind propagation delays". Please elaborate. Which technique is used and I assume this is from L1.
* * *
Response:

The technique used is described in this publication: Weimer, D. R. and J. H. King, Improved calculations of interplanetary magnetic field phase front angles and propagation time delays, J. Geophys. Res., 113, 2008. We will add the reference in the revision. This technique was referenced in all of the original, empirical model papers.

________________-

Referee #2:

L293. "a wrong result again, but useful to include." Can you please add an argument "a wrong result again, but useful to include, because. . ."
* * *
Response:

This revision will be included: ". . .a wrong result again, but useful to include as it shows

why the divergence-free component alone is not sufficient for calculating the Hall conductivity, as might be assumed. Showing these calculations are useful for understanding the contributions that both components have on the final numerical results."

————————————-

Referee #2:

L319. The Green et al paper was a proof of concept or technical paper.

—————————————

Response:

That is absolutely true, and this proof of concept paper had served as the inspiration for trying the technique using our models, for a wide range of conditions. While we had used the word "demonstrated," to describe the Green et al. paper, we will add "proof of concept" to the revision. While it was only a proof of concept, we still think it is useful to include the comparison in L319.

————————————-

Referee #2:

L339: See Ohtani, S., Gjerloev, J. W., Johnsen, M. G., Yamauchi, M., Brändström, U., & Lewis, A. M. (2019). Solar illumination dependence of the auroral electrojet intensity: Interplay between the solar zenith angle and dipole tilt. Journal of Geophysical Research: Space Physics, 124, 6636– 6653. https://doi.org/10.1029/2019JA026707.

—————————————

Response:

The results shown in Figure 6 do agree with the key point in the Ohtani et al paper, that "The nightside westward electrojet (WEJ) is more intense when the ionosphere is dark," so a reference can be added to the revision.

—————————-

Referee #2:

Clean up language. For example: "atmosphere with a high level of precision" what does this mean? "that cause the derived conductivity to seem negative", why seem? "gradients act to transport energy flux from", transport flux? "and potentially useful conductivity values are lost." to who and how?

—————————————

Response:

Thanks for pointing out that these sentences could have been worded better. The revised manuscript will include more clear language in the parts that are indicated.

---

## Author Comment (AC2) · 20 Oct 2020

Referee #1:

The paper presents a model of height integrated ionospheric conductance that is derived from results of other dedicated empirical models. The manuscript is well written, in most part clear and understandable and potentially suitable for publication at Annales Geophysicae.

There are few comments, that the authors shall addresses:

1. The model is a composite of different empirical models. Quantify the uncertainty

and/or error that accumulates by using the empirical models to build another one.

——————————-

Response:

The results are presented as maps that are calculated for various combinations of preset conditions, such as IMF magnitude, clock angle, and dipole tilt angle. However, the results were never directly referred to as a "model." We took care to not use that word to describe the results. Our view is that an empirical model takes any random combination of input parameters and outputs a result for specified locations and time. We have not created any such program, although the results could, in theory, be used to look-up a map file that most closely matches the desired conditions, as mentioned in response #3. Many more maps of the results are contained in the data archive than presented in the paper. As discussed in the response to item #2, we cannot obtain a good measurement of the uncertainty for each model. However, in comparison with existing conductivity models, our results are known to be within the the expected range of values, so the errors cannot be very great. More recently there have been two papers in the accepted or pre-print stage that have become available to us only after the submission of our manuscript. Our results are within range of the maps shown in these new papers, which provides confidence that our errors are not nearly as large as would be obtained by accumulating the uncertainties (in the range of 20%-50%) that result purely from the high-frequency fluctuations. Carrying out an accumulation of errors with uncertain input values which would be an exercise in futility, particularly due to the unknown influence of the uncertainties in the IMF values, which is discussed in the next point.

Citations of these new papers will be included in the revised paper:

Carter, J. A., S. E. Milan, L. J. Paxton, B. J. Anderson , and J. Gjerloev, Height-integrated ionospheric conductances parameterised by interplanetary magnetic field and substorm phase, J. Geophys. Res., doi:10.1029/2020JA028121 , 2020.

Mukhopadhyay, Agnit, Daniel T. Welling, Michael W. Liemohn, Aaron J. Ridley, Shibaji Chakraborty, and Brian J. Anderson, Conductance Model for Extreme Events: Impact of Auroral Conductance on Space Weather Forecasts, Space Weather, https://arxiv.org/pdf/2008.12276.pdf, 2020.

Liemohn, Michael W., The case for improving the Robinson formulas, J. Geophys. Res., doi:10.1029/2020JA028332, 2020.

————————————-

Referee #1:

2. This point relates to point 1. The Discussion mentioned a row of error that might be in the model results. Certainly, the small-scale variation cannot be captured, but this is not what would be expected from the empirical model and is quite clear. Also, as the major source of error, an uncertain solar wind measurement is mentioned. However, it needs to be checked how this is valid and that uncertainties from point 1, may not be more important.

————————————-

Response:

We need to make it more clear in a revision that "small-scale variation" includes high-frequency, temporary fluctuations, and such variations prevent an accurate measurement of the standard deviation in a global-scale, spatial mapping. As given in our response to Referee #2, the empirical models all use measurements that were taken as a function of time, while the satellites were moving in space, or in the case of the ground-based magnetometers, fixed in position while moving through local time. The original measurements contain fluctuations at various frequencies, with the most rapid fluctuations, lasting a few seconds or so, often having the largest magnitudes. The end-result models are global-scale, spatial representations of the fields that do not have the temporal fluctuations included. During the fitting process the difference between the

models and input values (the total square error) is minimized, and this error can be calculated. This is where the 20%-50% numbers come from. But since the high-frequency fluctuations are believed to be mostly temporal variations, rather than spatial, the least-square error that is calculated is not a true representation of the accuracy of the spatial models.

In the revision we will add a more detailed discussion about the differences between the IMF values that are measured by sentinels at the L1 orbit, and what impacts the Earth's magnetosphere. The issue is illustrated well in the 2018 paper by Borovsky that was already cited, that this referee is invited to inspect. Further evidence is given in a paper that was published online only recently, having a key point that "the solar wind measured by a single spacecraft at L1 often does not impact Earth in a homogeneous manner":

Burkholder, B. L., Nykyri, K., & Ma, X. (2020). Use of the L1 Constellation as a Multi-spacecraft Solar Wind Monitor. Journal of Geophysical Research: Space Physics, 125, e2020JA027978. https:// doi.org/10.1029/2020JA027978

As the submitted manuscript already indicated, the full, three-dimensional structure of the IMF cannot be determined unless additional sentinels are placed in orbit around L1. On the topic of the variable time delays we are adding these two references in response to Referee #2:

Weimer, D. R., D. M. Ober, N. C. Maynard, W. J. Burke, M. R. Collier, D. J. McComas, T. Nagai, and C. W. Smith. Variable time delays in the propagation of the interplanetary magnetic field. J. Geophys. Res., 107((A8)), 2002.

Weimer, D. R., and J. H. King. Improved calculations of interplanetary magnetic field phase front angles and propagation time delays. J. Geophys. Res., 113, 2008.

————————————-

Referee #1:

3. The conclusion repeats earlier finding from other authors and mention the shortcomings of the model results submitted here, but that the results might be of some value to the community. This again calls for a quantification, if rough, of the errors expected. To which application the value is expected?
* * *
Response:

The results could be used in AMIE-like assimilation models and numerical simulations of the coupled magnetosphere-ionosphere system. As we have not created an actual empirical model, the results would need to be used by plugging in our derived values for the IMF conditions that most closely match those being simulated, with interpolation and filling in of missing results. As done by Mukhopadhyay et al [2020], validation of the accuracy is obtained by testing whether or not there is an improvement in various metrics and skill scores in the predictions of ground-level magnetic perturbations. This type of test would be the best way to validate the results. Another possible test would be to feed the output of a FAC model or AMPERE measurements into a potential solver that uses the derived conductivity values, and compare the output with the electric potential patterns that are expected.
* * *
Referee #1:

4. L 23 says that earlier formulars are confusing and that the authors applied a simpler formula. What does justify the simplification?
* * *
Response:

Several versions of the conductivity formulas have been published, as cited in the paper. These mathematical equations contain different notations for the same calculations, but with identical results. Some formulas contain symbols for cyclotron frequencies, others use mobility coefficients, and some may explicitly include all of the terms, such as magnetic field strength, that go into the calculation of such quantities. In some texts the summation over different ion species is only implied. It is the differences between the various formulas, and their variations, that can be confusing, not the formulas themselves; the paper revision will need to state that more clearly. Equations (1) through (4) get to the same result as the others, but explicitly shows the summation over the different ion species; the simplification is just that (3) and (4) are used to simply formulas (1) and (2) by replacing multiple symbols with these two.

———————————-

Referee #1:

5. I had sometimes the impression that not the most recent developments of certain areas are referred to, such as for AMIE, SECS method, or NRLMSIS. Please check about newer developments in these fields.

———————————-

Response:

There have been too many variations of the AMIE and SECS methods to keep track of or cite them all, so the earliest or the most well-known papers are cited. The more recent NRLMSIS model, named simply NRLMSIS 2.0, is described in a paper that was only recently submitted and accepted (17 September 2020) so it could not be included in the first draft of this paper, but it can be referenced in the revision.

———————————-

Referee #1:

6. What is the opinion of the authors if direct observations near the E region would enhance their findings, such as is provided by a satellite mission like Daedalus?

(https://daedalus.earth)

—————————-

Response:

Daedalus would provide the opportunity to study more direct impacts of plasma flows in the E region, as the referee suggests. For example, this data would likely be interesting as a follow-up to an earlier work on the relationship between FAC and solar index work, particularly the FAC response to the solar indices representing the extreme ultraviolet emissions [Edwards, T. R., D. R. Weimer, W. K. Tobiska, and N. Olsen, Field-aligned current response to solar indices, J. Geophys. Res. Space Physics, 122:5798–5815, 2017]. While the Daedalus data would not have a direct impact on our conductivity calculations, it is possible that these data could confirm the existence of features shown in our Figures 6 through 9, and the influence that the IMF clock angle has on the conductivity distribution. Daedalus observations may perhaps be useful in future version of the IRI and/or NRLMSIS models.